# Probiotic Mechanisms Affecting Glucose Homeostasis: A Scoping Review

**DOI:** 10.3390/life12081187

**Published:** 2022-08-03

**Authors:** Maša Pintarič, Tomaž Langerholc

**Affiliations:** Department of Microbiology, Biochemistry, Molecular Biology and Biotechnology, Faculty of Agriculture and Life Sciences, University of Maribor, Pivola 10, 2311 Hoče, Slovenia; tomaz.langerholc@um.si

**Keywords:** probiotic, mechanism, glucose, metabolism, homeostasis

## Abstract

The maintenance of a healthy status depends on the coexistence between the host organism and the microbiota. Early studies have already focused on the nutritional properties of probiotics, which may also contribute to the structural changes in the gut microbiota, thereby affecting host metabolism and homeostasis. Maintaining homeostasis in the body is therefore crucial and is reflected at all levels, including that of glucose, a simple sugar molecule that is an essential fuel for normal cellular function. Despite numerous clinical studies that have shown the effect of various probiotics on glucose and its homeostasis, knowledge about the exact function of their mechanism is still scarce. The aim of our review was to select in vivo and in vitro studies in English published in the last eleven years dealing with the effects of probiotics on glucose metabolism and its homeostasis. In this context, diverse probiotic effects at different organ levels were highlighted, summarizing their potential mechanisms to influence glucose metabolism and its homeostasis. Variations in results due to different methodological approaches were discussed, as well as limitations, especially in in vivo studies. Further studies on the interactions between probiotics, host microorganisms and their immunity are needed.

## 1. Introduction

Glucose is one of the simple sugars (monosaccharides) and as such it represents the main fuel for the normal functioning of the cell [1]. Maintaining of glucose homeostasis [2] is therefore essential to prevent the development of diseases associated with a disturbance of glucose homeostasis, such as insulin resistance [3], glucose intolerance [4] and type 2 diabetes [5]. The pathological condition common to all three is the occurrence of an elevated blood glucose level, i.e., hyperglycaemia [6]. The condition of persistent hyperglycaemia is very dangerous and leads to more serious complications associated with heart disease, eye, kidney and nerve damage [7], as well as with cancer [8]. Although the aetiology of hyperglycaemia has not been clearly understood, the main causes are attributed to genetic nature and/or to environmental factors [9,10]. The latter include an unhealthy lifestyle such as poor diet, obesity and lack of physical activity [11,12].

Over the last decade, most studies on the aetiology and development of metabolic diseases have indicated the increasingly important role of the gut microbiota. It is now known that dysbiosis of the microbiota is associated with significant metabolic consequences leading to abnormality of physiological processes in the body [13]. In fact, dysbiosis leads to increased permeability of the intestinal wall, resulting in elevated levels of bacterial endotoxins, (i.e., lipopolysaccharides - LPS) in the blood, which in turn can cause inflammation. Persistent inflammation can eventually lead to pathological metabolic conditions such as obesity, insulin resistance [14,15], metabolic syndrome [16,17], type 2 diabetes [18,19,20,21] and hyperglycaemia [22,23].

Selected microbial strains commonly known as probiotics have been documented to have a beneficial effect on glycaemic control in the blood (reviewed in [13,24,25,26]). In addition, there is growing evidence that microorganisms play an important role in glucose homeostasis, particularly in metabolic conditions such as obesity and obesity-induced insulin resistance [14,15], metabolic syndrome [16,17], type 2 diabetes [18,19,20,21,27] and cardiovascular disorders [23,28]. Despite different opinions on the terminology of probiotics [29], a microorganism can be classified as a probiotic if it meets most of the safety, functional, technological and physiological criteria [30,31]. Most probiotics derive from a heterogeneous group of lactic acid bacteria, dominated by bacteria from genus *Lactobacillus* [32], *Lactococcus*, *Enterococcus* and *Streptococcus* predominate [33]. Probiotics also include strains of other bacterial species, (i.e., *Bifidobacterium* sp. [34], *Bacillus* sp., *Propionibacterium* sp., *Escherichia coli*), yeasts (*Saccharomyces*) and moulds (*Aspergillus*) [33,35]. Although probiotics are “live microorganisms that when administered in adequate amounts confer a health benefit on the host” [29], their non-viable components named paraprobiotics may be a safer yet effective alternative for use in vulnerable individuals [36].

Despite the existence of several reviews mentioned above, knowledge of mechanistic studies on the beneficial effects of probiotics in glucose-related pathological conditions is scarce. Therefore, the aim of the review was to evaluate the most recent literature (between 2010 and 2021), in which probiotic administration has direct or indirect effects on glucose metabolism and its homeostasis. The review has been divided into sections regarding probiotic effects on the physiology of different human and animal organs. Concluding remarks and future perspectives on the importance of probiotics and their influence on glucose-related pathological disorders have been proposed.

## 2. Materials and Methods

### Search Strategy and Studies’ Selection

Identification of the scientific literature accessible in English was performed using a combination of search terms (“probiotics; mechanisms; glucose”) in a publication period between January 2010 and December 2021. Studies were identified using the PubMed (https://pubmed.ncbi.nlm.nih.gov/, accessed on December 2021), ScienceDirect (https://www.sciencedirect.com/, accessed on December 2021) and Scopus (https://www.scopus.com/search/form.uri?display=basic#basic, accessed on December 2021) databases. The preliminary search yielded a total of 7079 records. Duplicates were excluded, as well as abstracts, due to insufficient available data. Further screening met the inclusion criteria, namely in vivo (human and animal) and in vitro studies showing any direct or indirect effect of probiotics (alone or in combination with other probiotics) on glucose, glucose metabolism or homeostasis. After considering all of the above criteria, 137 final studies were included in this review. The obtained data from the publications were further sorted based on the reported outcomes, (i.e., the different probiotic effects).

## 3. Effect of Probiotics on Glucose Metabolism and Homeostasis

The incomplete understanding of how probiotics work is due to a different methodological approach, a lack of investigation into mechanisms and major physiological differences between probiotic strains [37,38], which makes it difficult to comprehend the significance of their use, especially in glucose-related pathological conditions.

### 3.1. General Remarks about the Mechanisms of Probiotics

The qualitative and quantitative composition of the gut microbiota has a major impact on the interaction with the host. The current strategy for treating dysbiosis is still largely based on the use of probiotics to restore microbial diversity and normalise the “disturbed” gut microbiota [39]. Microbiota acts at different levels through complex mechanisms that are not fully understood [40]. As a result, studies of the potential mechanistic actions of probiotics often encounter a number of limitations:
The use of oversimplified in vitro models that often fail to reproduce the results in vivo;The use of “human” probiotics in animal models in vivo, (e.g., rodents), which do not take into account the functional significance of the species- and strain-specific administration of probiotics and their impact on the host’s immune response and its microbiota;Probiotics’ action ultimately requires the involvement of the endogenous microbiota, which is host-specific and often diet-dependent and therefore hardly reproducible;Most of the mechanisms of probiotics presented in research studies have suggested two main principles of probiotics’ action [41];Direct, contact-dependent principle (binding to different surface molecules);Indirect principle via secretory molecules (production of bioactive peptides and metabolites).

### 3.2. Effect of Probiotics on Blood Parameters

#### 3.2.1. In Vivo Human Studies

A number of human clinical trials demonstrated differential effects of probiotics on blood parameters in patients with various metabolic problems. These parameters reflect a direct or indirect relationship with glucose and its concentration in the blood (glycaemia) (Table 1).

Several studies have measured various blood parameters related to glycaemic status that may indicate possible control of blood glucose levels (glycaemic control). These can be divided into blood parameters, which are directly related to glycaemic status, such as fasting blood glucose, or blood parameters, which are indirectly related to glycaemic status, such as insulin, glycohaemoglobin (HbA1c) and homeostasis model assessment-estimated insulin resistance index (HOM-IR). However, some authors have included additional blood parameters, indirectly related to glycaemic status, in their studies, which are explained in more detail in Table 1. Tonucci et al. [50] have shown that glycaemic control is improved by taking probiotic fermented milk for 6 weeks. In one trial, administration of a probiotic preparation containing *Lactobacillus casei* Shirota [56] after 4 weeks maintained glycaemic control and preserved insulin levels in fasting state. A 6-month human clinical trial also reported beneficial effects of two probiotics (*Lactobacillus reuteri* ADR-1 and ADR-3) on patients with type 2 diabetes. ADR-3 reduced some parameters of glycaemic control, while ADR-1 had a beneficial effect on lowering blood pressure [51]. In another 12-week human clinical trial, probiotic did not alter HbA1c levels, fat levels or total serum bile acids [54]. Furthermore, Shellekens et al. [45] have demonstrated some anti-obesity effects of *Bifidobacterium longum* APC1472 in humans, possibly due to an alternation in ghrelin signalling. Since the gut hormone ghrelin may be involved in glucose homeostasis via inhibition of insulin secretion [57], administration of *Bifidobacterium longum* APC1472 may be clinically significant in patients with pre-diabetes and type 2 diabetes mellitus. In addition, the authors speculated that the reduction in fasting cortisol levels due to probiotic administration affects the hypothalamic–pituitary–adrenal (HPA) axis, which regulates the pancreas and simultaneously influences glucagon and insulin secretion. All of these effects may contribute to glucose level reduction. A 12-week administration of *Lactobacillus plantarum* OLL2712 in pre-diabetic patients has shown improvement in fasting plasma glucose levels, glycoalbumin levels and insulin resistance [46]. In a trial in healthy adults, administration of the paraprobiotic *Lacticaseibacillus casei* 01 resulted in a better reduction in blood glucose levels than the probiotic of the same name [42].

The HOM-IR index is an important parameter in the context of cardiological and metabolic disorders, as it contributes to the early detection of insulin resistance and to the assessment of the risk of developing diabetes, cardiovascular pathologies and atherosclerosis [58]. The effect of probiotics on blood parameters was also demonstrated in a clinical trial conducted in the Arab population [47], in which a 6-month intake of a multi-strain probiotic supplement (Ecologic^®^ Barrier) showed a decrease in the HOM-IR index as well as a decrease in the level of endotoxin and inflammatory adipokines. The authors suggested the use of the probiotic supplement as a dietary supplement for patients with type 2 diabetes, but also emphasised the wide variation in results due to genetic diversity, dietary habits and environmental differences (geographical regions) between the populations studied [47]. Kobyliak et al. [52] also demonstrated the impact of a multi-strain probiotic (“Symbiter”) on reducing HOM-IR. Similar results were reported by Khalili et al. with *Lactobacillus casei* 01 [49] and *Lactobacillus casei* [48] as a supplement. In patients with metabolic syndrome, a two-month intake of probiotic yoghurt (*Lactobacillus acidophilus* La5 and *Bifidobacterium lactis* Bb12) led to a reduction in blood glucose levels and significant changes in insulin resistance (HOMA-IR) and sensitivity. The authors therefore believe that regular consumption of probiotic yoghurt has a positive effect on the treatment of metabolic syndrome [44].

Interleukin-6 (IL-6), together with monocytochemotactic protein-1 (MCP-1), is a pro-inflammatory cytokine associated with the development of insulin resistance and hyperglycaemia [52]. In addition, a pro-inflammatory interleukin 1 beta (IL-1β) cytokine may be involved in postprandial inflammation and thus in the regulation of glucose homeostasis and immune response [59], while tumour necrosis factor alpha (TNF-α) seems to be mainly involved in insulin signalling and action [60] and in the regulation of glucose and lipid metabolism [61]. However, Kobylak et al. additionally showed that glycaemia-related parameters such as body weight, body mass index (BMI) and IL-6 remained unchanged or did not significantly decrease (HbA1c) after administration of multi-strain probiotics. However, the same compounded probiotics altered some pro-inflammatory factors such as TNF-α and IL-1β [52]. Toshimitsu et al. [46] have shown that both parameters as well as IL-6 were suppressed after 12 weeks of treatment with *Lactobacillus plantarum* OLL2712 in a pilot trial with pre-diabetic patients.

HbA1c is an important parameter in the long-term monitoring of blood glucose levels (balance) and indicates the number of glucose molecules bound to haemoglobin in the red blood cells (erythrocytes) [62]. The mechanism by which probiotics reduce the amount of this parameter is still unclear [26]. Five clinical studies [43,50,51,52,53] showed a statistically significant reduction in HbA1c levels after taking a probiotic preparation, while two other studies found no change in the amount of this parameter [49,63].

Fetuin-A is known as a glycoprotein, which is secreted by liver and adipose tissue. Its increase in the blood is indirectly related to patients with atherosclerosis, insulin resistance, diabetes mellitus and metabolic syndrome [64], as it affects the activity of insulin receptor tyrosine kinase [65]. Sirtuins (SIRTs) are very important regulators of energy homeostasis and metabolism due to their deacetylation activity in variety of organs [66]. Fetuin-A and SIRTs were investigated in a randomised controlled trial in patients with type 2 diabetes after an 8-week intake of *Lactobacillus casei* 01 and *Lactobacillus casei* [48,49]. The results of blood analysis showed that fetuin-A levels were decreased, while SIRTs levels were increased.

Despite the proven beneficial effects of probiotic therapy in patients with type 2 diabetes, some studies failed to demonstrate changes in blood and metabolic parameters or the changes were not statistically significant, which is a crucial justification for a more objective approach and evaluation of the effectiveness of probiotic therapy. The diversity of results is mostly a consequence of the different methodological approaches in the studies: use of one or more probiotic strains, their amount and duration of administration and the number of patients. Moreover, it is already known that the activity of probiotics is species-specific [67]. A comparison of the different results of the individual clinical probiotic studies can be carried out within the framework of meta-analytical studies. In this way, a more objective view of the potentially beneficial effect of probiotics on various blood and metabolic parameters in patients with type 2 diabetes can be obtained [67,68,69,70,71,72,73,74,75,76]. However, the above meta-analyses point to numerous limitations, even if they ultimately confirm the multiple beneficial effects of probiotics on blood and metabolic parameters: small groups of study participants [67,68,69,71,74], lack of data on dietary habits and physical activity of study participants [68,70,73], diversity within a study [69,71,72,73,74], short duration of probiotic therapy, different therapeutic doses, use of different probiotic strains [67,68,70,72,74], lack of evidence on possible side-effects of probiotic use [69] and lack of testing of other metabolic factors affecting glycaemic control [70,71]. Last, but not least, Tao et al. [76] proposed that more clinical data and research on the mechanisms of probiotic are needed.

#### 3.2.2. In Vivo Animal Models Studies

Many studies in animal models demonstrated positive effects of various probiotics on diverse blood parameters, which are summarised in Table 2. The range of measured parameters was similar to in vivo human studies showing potential effects of probiotics on improving glucose tolerance, insulin resistance and insulin sensitivity.

Oral intake of probiotic preparations of the strains *Lactobacillus casei* and *Bifidobacterium bifidum* (alone or both simultaneously) reduced the incidence of hyperglycaemia (increased blood glucose concentration) and dyslipidaemia (imbalance of blood lipid concentration) in rats with induced diabetes [79]. Similar antidiabetic effects in rats were obtained by administration of probiotic strains *Lactobacillus plantarum* TN627 [80], *Bifidobacterium animalis* 01 [81], *Lactobacillus paracasei* HII01 [82], 9 strains of *Lactobacillus rhamnosus*, *Bifidobacterium adolescentis* and *Bifidobacterium bifidum* [77], *Lactobacillus rhamnosus* BSL and R23 [83], heat-inactivated *Streptococcus thermophilus* [84], *Lactiplantibacillus plantarum* IMC 510 [85] and *Lactobacillus fermentum* RS-2 [86]. In the latter case, in addition to reduced glucose levels in fasting state, a low oxidative stress was observed [86].

Studies in mice and rats have shown, among other things, a positive effect of probiotic preparations on glucose balance [78,87,88,89,90,91,92,93,94,95,96,112,113,118] and reduced insulin resistance [78,87,89,90,91,92,94,113]. Andersson et al. [97], Sakai et al. [98], Park et al. [114], Naito et al. [99], Wang et al. [100,101], Machado et al. [102], Lee et al. [103], Hsu et al. [115], Kim et al. [116] and Zeng et al. [104] have shown that the probiotic addition of different species of the genus *Lactobacillus* or a combination of probiotics [100,101,116] improves glucose sensitivity (tolerance) [97,98,99,100,101,102,103,114,115,116] and reduces insulin resistance [97,98,99,100,101,114] by decreasing serum glucose levels in obese/diabetic mice and rat models. Endotoxaemia, along with inflammation, is a common factor in metabolic diseases (especially obesity, insulin resistance and diabetes). In a study conducted in mice, the addition of *Lactobacillus casei* Shirota reduced insulin resistance and endotoxaemia. In this context, it has been demonstrated that the probiotic effect on glucose regulation is not necessarily directly related to the body weight and the amount of fat in the animals [99].

Effects on glucose homeostasis and cholesterol metabolism have also been demonstrated by Ashrafian et al. [105], Wang et al. [106], Kim et al. [116], Tarrah et al. [107] and Sun et al. [108]. In the latter study in mice, *Lactobacillus acidophilus* SJLH001 had a statistically significant effect on the regulation of transcription genes, essential for glucose transport and cholesterol metabolism [108]. Yan et al. [110] have demonstrated a beneficial effect of *Lactobacillus acidophilus* on type 2 diabetes in mice by controlling liver glucose levels, lipid metabolism and gut microbiota.

Furthermore, in a study on mice [78,91,94,113,117] and rats [84,89], serum levels of certain inflammatory markers such as IL-6, TNF-α and IL-1β were decreased after a specific probiotic administration. In addition, heat-killed *Streptococcus thermophilus* [84], *Bifidobacterium animalis* 01 [81] and *Lactobacillus casei* LC89 [91], *Lactobacillus plantarum* CCFM0236 [78] increased the levels of IL-10, an important anti-inflammatory marker, when administered to rats and mice with type 2 diabetes, respectively.

A study in rats [81] showed a statistically significant reduction in HbA1c levels after taking probiotic supplements. In addition to ageing and neurodegenerative diseases, impaired memory function in the brain may also be associated with impaired glucose metabolism, i.e., an increase in plasma HbA1c levels [117]. Moreover, Bonfili et al. have demonstrated the beneficial effects of the probiotic formulation SLAB51 in a mouse model of Alzheimer’s disease. The probiotic formulation lowered serum HbA1c levels, showing a positive effect on glucose homeostasis [111].

Osteocalcin is a protein hormone produced by osteoblasts. Primarily it inhibits bone formation, but as a hormone it plays an important role in the synthesis of testosterone in testis as well as synthesis of the muscle mass [119]. However, it has been proven to have a positive effect on improving glucose tolerance and regulating glucose metabolism [119,120,121]. The addition of *Lactobacillus casei* Zhang in hyperinsulinaemic rats increased the amount of osteocalcin, which in turn influenced the improvement of glucose tolerance. The regulation of glucose tolerance is probably induced via the osteocalcin–adiponectin pathway [109].

#### 3.2.3. In Vitro Studies

The enzyme alpha-glucosidase (α-glucosidase) catalyses the digestive processes of carbohydrates and is responsible for increased blood glucose levels [118]. In in vitro studies [78,94,108,118,122], certain probiotics inhibited α-glucosidase activity and thereby decreasing the level of glucose ready for absorption and thereby glycaemic index of food.

### 3.3. Effect of Probiotics on Brain Function

The apparent communication between the intestinal microbiota and the central nervous system via the “gut-brain axis” has led researchers and/or clinicians to investigate the preventive and therapeutic use of probiotics in various neurological disorders [123].

Recent studies have revealed an important role of glucose homeostasis in some brain pathological conditions, such as Alzheimer’s disease (AD) [117]. Bonfili et al. have shown that the administration of probiotic SLAB51 increased glucose uptake in mice with Alzheimer’s disease due to the restoration of glucose receptor 3 (GLUT3) and glucose receptor 1 (GLUT1), the key glucose receptors in brain. The increased expression of both glucose receptors was in line with alleviated levels of phosphorylated 5′ adenosine monophosphate-activated protein kinase (AMPK) and protein kinase B (AKT), which are important metabolic regulators in most cell types. Moreover, they reported an increased expression of brain insulin-like growth factor receptor β (IGF-IRβ). IGF-IRβ is considered as a promising therapeutic target in AD, because of its important part in proteasome-mediated removal of oxidised proteins [111].

### 3.4. Effect of Probiotics on Bile Acid Metabolism

Bile acids are highly effective surface-active digestive substances (surfactants) and natural emulsifiers whose main function is to support the digestion and absorption of fat-soluble lipids and vitamins. Their cholesterol production takes place exclusively in the hepatocytes. After they have fulfilled this function in the small intestine, they are reabsorbed into the bloodstream as part of the so-called “enterohepatic recirculation”. From here, their path leads back to the liver, where their activity is related to the various signalling pathways [124].

Probiotics containing bile salt hydrolase (BSH) have been shown to interfere with the metabolism of bile acids and promote the deconjugation of bile salts via BSH, which is important in chronic metabolic diseases. In human studies, daily addition of *Lactobacillus reuteri* NCIMB 30242 increased the amount of free bile acids that were no longer absorbed in the intestinal wall as a result of probiotic BSH deconjugation, but were excreted in the faeces. It was also speculated that deconjugation might be directly related to the increase in fibroblast growth factor 19 (FGF) [125]. In fact, FGF plays an important role in the regulation of bile acid synthesis and its amount directly influences lipid and glucose metabolism [126].

In a human clinical study, *Lactobacillus reuteri* DSM 17938 can contribute to an increase in the level of deconjugated bile acids, which was also evidenced by the positive correlation with an increase in insulin sensitivity and thus an improvement in glucose metabolism. The authors hypothesised that the probiotic influences the biosynthesis of deconjugated bile acids by regulating genes for certain crucial enzymes involved in the biosynthesis [54].

In a study in rats the probiotic *Lactobacillus casei* Zhang [109] increased gene expression of liver X-receptor-α (LXR), which has already been shown to increase glucose tolerance and cholesterol reduction via increased BSH levels and deconjugation of bile acids [127]. In another study conducted in rats [128], the same probiotic influenced the reduction of bacteria with hydrolase activity, resulting in increased bile acid secretion.

### 3.5. Effect of Probiotics on Adipose Tissue Function and Inflammation

For many years, adipose tissue was considered to be an inactive organ with a single function, the storage of lipids. However, a number of studies have shown that it is an active organ, which plays an essential metabolic and hormonal role in homeostasis of energy metabolism and glycaemic regulation [16,129] (Figure 1). Excessive accumulation of lipids in the adipose tissue leads to its hypertrophy, cellular stress and local inflammation. The latter is chronically associated with obesity, insulin resistance, hyperglycaemia and type 2 diabetes [130].

In studies on high-fat diet fed mice, some probiotic strains have demonstrated their protective role against chronic inflammation in adipose tissue, leading either to its prevention or its inhibition by reducing insulin resistance and hyperglycaemic state. Indeed, a high-fat diet has been shown to increase some of the pro-inflammatory factors such as fat TNF-α, IL-6, IL-1β. *Lactobacillus fermentum* MTC5689 decreased the expression of pro-inflammatory indicator genes, TNF-α and IL-6 in visceral adipose tissue of mice [131]. In the second study in mice, *Lactobacillus sakei* OC67 reduced the incidence of hyperglycaemia and obesity by lowering inflammation [132]. In both studies, as well as in three other studies [98,133,134] reductions in glucose levels in fasting state and a decrease in IL-1β expression have been reported. The reduction of TNF-α expression in fat has also been shown after the addition of *Lactobacillus fermentum* LM1016 [133], *Lactobacillus crustorum* MN047 and *Lactobacillus rhamnosus* LS-8 [134]. The later probiotic decreased the IL-6 expression in adipose tissue as well [134]. The effect of *Lactobacillus rhamnosus* GG also showed the improvement and reduction of glucose hypersensitivity and insulin resistance in obese mice. The addition of the probiotic was able to reduce the expression of genes for specific macrophage factors important for macrophage infiltration and activation, leading to a reduction in macrophage activation, inflammation and thus improved insulin sensitivity in adipose tissue [114].

Deng et al. [135] suggested that different genotypes of *Akkermansia municiphila* exert specific effects in repairing damage to adipose tissue caused by a high-fat diet in mice. The study showed that different genotypes of probiotic can play different roles in the same disease state, particularly in combating endotoxaemia, inflammation and whitening of brown adipose tissue, glucose tolerance, hyperlipidaemia and hepatic steatosis.

Strains of the genera *Bifidobacterium* [136] and *Bacillus* [113] have been shown to influence the increased expression of proteins involved in the insulin signalling pathway in mice. Adiponectin, along with leptins, are involved in controlling feeding behaviour and increasing insulin sensitivity [137]. Increased adiponectin mRNA levels in fat cells (adipocytes) [136] and increased serum leptin levels [113] have also been demonstrated in the above studies. Due to the increased insulin sensitivity, probiotics influenced the improvement of glucose absorption in tissues by elevating adiponectin and leptin concentrations, which led to a decrease in blood glucose levels [138]. However, in a study in mice [139], adiponectin blood levels as well as its mRNA levels in adipose tissue were decreased and restored by the administration of probiotic supernatants.

Peroxisome proliferator-activated receptor α (PPARα) is important for processes controlling energy expenditure, inflammation and glucose homeostasis [140]. Due to increased levels of a hepatokine mediator of adiponectin expression in the regulation of glucose and lipid metabolism-fibroblast growth factor 21 (FGF21), increased PPARα mRNA levels and butyric acid levels, the authors suggested a possible link between probiotic mechanism-butyric acid-PPARα-FGF21-adiponectin-glucose and lipid metabolism [139]. Indeed, Molina-Tijeras et al. [141] linked the improvement in glucose and lipid metabolism to the upregulation of PPARα expression after administration of *Lactobacillus fermentum* CECT5716.

The downregulation of *sirt1* (sirtuin-silent mating type information regulation 2 homolog 1) gene expression is an indicator of metabolic dysregulation in many organs, including adipose tissue. In a study on mice, *Lactobacillus plantarum* increased adiponectin levels, which may have led to an upregulation of *sirt1* gene expression and thus to an improvement in lipid metabolic dysfunction [142].

The role of the major adiponectin receptor gene, i.e., adiponectin receptor 2 (AdipoR2), has been linked to insulin resistance and type 2 diabetes, where it plays a crucial role in glucose and lipid metabolism, inflammation and oxidative stress [143]. In a study on rats [109], the probiotic *Lactobacillus casei* Zhang also increased the expression of AdipoR2 in fat cells. The same probiotic has also increased the expression of the genes for the glucose receptor 4 (GLUT4) and core receptor-peroxisome proliferator-activated receptor gamma (PPAR-γ). Both parameters consequently had a positive effect on oral glucose tolerance. PPAR-γ is a known receptor that maintains the expression of key glucose regulatory and liporegulatory molecules, important for insulin signal transduction [144], while GLUT4 is the major glucose transporter for the facilitated transfer of glucose from blood to skeletal fat tissue and muscle [145]. A possible link between improved glycaemic levels, utilisation of glucose and increased expression of *GLUT4* genes following administration of *Lactobacillus fermentum* CECT5716 has also been proposed [141]. In addition to PPAR-γ, some probiotics have also been shown to affect increased expression of PPAR-α. in animal models [105,146] and in one in vitro study [147]

A transforming growth factor beta (TGF-β) is known to be an important parameter in adipose tissue due to its effects on inflammatory mediators, energy homeostasis, fat expansion and collagen deposition [148,149]. A decrease in TGF-β mRNA levels after administration of *Akkermansia muciniphila* as well as its extracellular vesicles (EVs) suggested a potential antidiabetic and anti-obesity effect of both the bacterium and its EVs [105].

*Lactobacillus plantarum* Ln4 significantly reduced lipid deposition and induced glucose absorption in 3T3-L1 adipocytes in an in vitro study [150].

### 3.6. Effect of Probiotics on Skeletal Muscle

The endoplasmic reticulum (ER) plays a role as a sensor for changes in the homeostasis of a cell function. Stress therefore disrupts the normal function of the ER, which in turn reacts differently to protect the cell. It is known that lipid metabolism and ER stress are closely linked in skeletal muscle cells. In addition, ER stress in muscle cells is also a major cause of insulin resistance, which in turn affects glucose homeostasis [151]. In a model study in mice, certain strains of *Lactobacillus* species have been shown to influence the reduction of ER stressors in skeletal muscle, thereby reducing lipotoxicity. In fact, the probiotic effect showed a decrease in the expression of certain genes that can alter lipid metabolism [114,131].

Administration of *Lactobacillus casei* or *Bifidobacterium bifidum* strains or a combination of both significantly increased the amount of muscle glycogen in rats with induced diabetes, which had a direct effect on reducing hyperglycaemia [79]. Similarly, Liu et al. showed an increase in protein levels of some glucose homeostasis-related molecules such as phosphoinositide 3-kinase (PI3K), phosphorylated protein kinase B (p-AKT) and glycogen synthase kinase 3 beta (GSK-3β). The latter is directly related to increased glycogen production [139].

The protective effect of *Lactobacillus plantarum* was demonstrated in a study conducted on mice [142]. The expression of irisin, a specific adipomyokine produced by skeletal muscle and adipose tissue, was upregulated, which in turn increased the insulin sensitivity of skeletal muscle and improved hepatic glucose and lipid metabolism. Since irisin is a peroxisome proliferator-activated receptor-γ-coactivator (PGC)-1α-dependent molecule, the authors speculated on a possible concurrent probiotic mechanism for improving metabolism based on the regulation of mitochondrial biogenesis.

Kim et al. [152] have also demonstrated the effect of a probiotic on increased expression of genes relevant to glycogen synthesis (glycogen synthesis-related gene pp-1) and increased expression of GLUT4 in skeletal muscle. In the same in vitro study, the authors showed improved insulin-dependent glucose uptake and increased expression of the GLUT4 and PPAR-γ genes.

The increase in GLUT4 expression due to the effect of the probiotic additive has been demonstrated in three animal studies: in mice [153] and rats [82,146]. Toejing et al. [82] linked the activation of GLUT4 translocation to the membrane in skeletal muscles to AMPK activation, which consequently affects the glucose uptake. The effects of probiotics on skeletal muscle in relation to glucose homeostasis and metabolism are shown in Figure 2.

### 3.7. Effect of Probiotics on Liver, Pancreas and Kidney

A pathological state of diabetes reduces the antioxidant potential of pancreatic and hepatic tissue and consequently increases the harmful effects of free radicals [154]. Studies on animal models have shown that single probiotics reduce oxidative stress in the pancreas [79,104,155], liver [90,93,104,156] and kidney [93,104], thus helping to increase the antioxidant activity and physiological function of the organ.

#### 3.7.1. Liver

Figure 3 shows some crucial effects of selected probiotics on the liver that lead to a change in glucose metabolism.

Increased liver glycogen levels and decreased blood glucose levels after supplementation with *Bacillus toyonensis* SAU-19 in high-fat diet/streptozicin-induced diabetic mice indicated a probiotic effect on glycogen synthesis [90]. Some parameters of liver metabolism, including liver glucose decreased after administration of *Lactobacillus plantarum* wild type strain (*δplnEFI*) in a study on diet-induced obese mice [157].

The addition of *Bifidobacterium lactis* HY8101 [156] and *Lactobacillus fermentum* LM1016 [133] to mice decreased the expression of genes relevant to the regulation of hepatic gluconeogenesis, i.e., phosphoenolpyruvate carboxykinase 1 (*PCK1*) and glucose-6-phosphatase catalytic subunit (*G6PC*), resulting in a reduction of blood glucose levels in fasting state. The same parameters, as well as β-cell analysis, were studied after administration of *Lactobacillus plantarum* HAC01 [103] and a multi-strain probiotic supplement Probioglu^TM^ [89] in diabetic mice. In both studies, the probiotics had a protective and restorative effect on β-cells. Improvement in glucose tolerance and insulin sensitivity due to the downregulation of G6PC [83,158] and phosphoenolpyruvate carboxykinase (PEPCK) [158] was also reported in diabetic rats by Farida et al. [83] and by Huang et al. [158] in olanzapine-induced diabetic mice, when they administered *Lactobacillus rhamnosus* BSL/*Lactobacillus rhamnosus* R23 and *Akkermansia muciniphila*, respectively. For both parameters, similar results were also observed by Okyere et al. after administration of *Bacillus toyonensis* SAU-19 [90]. Moreover, an improvement in glucose metabolism has been demonstrated by the upregulation of genes related to glycogenesis and glucose transport, such as Forehead Box O1 (*FOXO1*), glycogen synthase (*GS*), phosphofructokinase-1 (*PFK-1*) and glucose transporter 2 (*GLUT2*), respectively.

Park et al. [159] decreased gene expression of the enzyme PEPCK, which is critical for hepatic gluconeogenesis, by supplementing high-fat diet-fed mice with *Lactobacillus acidophilus* NS1 (NS1). The reduced expression of the enzyme was also confirmed in vitro by demonstrating that NS1 regulates reduction by activating transcription of hepatocyte nuclear factor 4 alpha (HNF4α), a core receptor and a key element in hepatic differentiation and cell proliferation [160].

Glucagon is an important hormone for maintaining blood glucose homeostasis. It regulates the activation of certain molecules in its pathway, such as the heterotrimeric Gs protein alpha-subunit (Gnas), cAmp-dependent protein kinase (PKA), CREB-regulated transcriptional coactivator 2 (CRTC2), as well as PEPCK and G6Pase [161,162]. Zhang et al. have demonstrated a probiotic effect on the glucagon signalling pathway in the liver. *Lactobacillus casei* LC89 downregulated all of the above molecules in the glucagon signalling pathway. Moreover, a reduction in blood glucose and an increase in liver glycogen have been demonstrated. Both parameters are also closely related to the downregulated genes of the above molecules [91].

The renin-angiotensin system (RAS) is a well-defined regulator of blood volume (its pressure) and total vascular resistance [163]. One of the major components of RAS is the angiotensin-converting enzyme 2/angiotensin-(1–7)/mitochondrial assembly receptor (ACE2/Ang-(1–7)/MasR) axis. Once activated, this axis plays a crucial role in many organs, especially in the liver, where it has already shown its potential as a regulator of many metabolic disorders [164,165]. Machado et al. [102] has demonstrated the potential effect of *Bifidobacterium longum* administration on improving metabolism in the liver of obese mice by increasing the expression of hepatic ACE2 and MASR. The increased expression of both parameters is directly related to the increase in Ang-(1–7), an important vasodilating agent that can reduce gluconeogenesis in the liver and improve glucose and lipid metabolism [166].

In two studies conducted in mice, *Lactobacillus paracasei* TD062 and *Lactobacillus plantarum* HAC01 also reduced the expression of certain glucose-linked genes (PEPCK) and had a positive effect on the PI3K/AKT signalling pathway, which is crucial for the biological activity of insulin and glucose uptake [103,122]. A similar effect was observed in a rat study [81] after administration of *Bifidobacterium animalis* 01. However, Wang et al. [100] have demonstrated a putative link between the protective effect of probiotics on pancreatic apoptosis and the upregulation of the PI3K/AKT pathway. In addition, probiotic administration showed protective properties against oxidative damage by affecting Kelch-like ECH-associated protein 1/nuclear factor-erythroid factor 2-related factor 2 (Keap1/Nrf2). The latter is important for the regulation of antioxidant genes [81]. The downregulation of Nrf2 expression associated with oxidative stress in the liver was also demonstrated by Wang et al. when they administered *Bacillus amyloliquefaciens* SC06 to mice fed a high-fat diet [113]. A positive effect on insulin sensitivity was also observed in mice after administration of *Lactobacillus rhamnosus* GG supernatant [139] and two strains of *Lactobacillus acidophilus* [110] with significant increases in PI3K [139] and phosphorylated (p)-AKT [110,139] as well as GSK-3β [110,139].

In high-fat diet and streptozoticin-induced diabetes mice, five different probiotic strains out of nine tested [77], two different *Lactobacillus acidophilus* strains [110] and *Bacillus toyonensis* SAU-19 [90] reduced TNF-α [77,90,110], IL-1β [77,90,110] and IL-6 [77,110] in liver homogenates pointing out their anti-inflammatory effect, which is also consistent with their hypoglycaemic effect, i.e., reduction in serum glucose levels [77,90].

Three probiotic strains, *Lactobacillus paracasei* 1F-20, *Lactobacillus fermentum* F40-4 and *Bifidobacterium animalis* subsp. *lactis* F1-7, were selected to test their effect on glucose uptake in HepG2 cells (hepatocytes), leading to their enhancement and thus promoting glucose metabolism in the liver [147].

However, the efficacy and safety of the use of probiotics in liver is still the subject of research, which is why Stavropoulou and Bezirtzoglou [123] do not yet recommend their use for the treatment of most liver diseases.

#### 3.7.2. Pancreas

Some in vitro studies and studies on experimental animals have shown the effect of probiotics on preventing damage to pancreatic tissue by preventing degeneration and inflammation of pancreatic islets or inducing their regeneration. Therefore, the addition of probiotics is expected to improve glucose tolerance [77,78,91,96,118,156,167]. In addition, Wang et al. [100,101] have shown a significant improvement in pancreatic morphology and insulin secretion as a result of probiotic administration. In a study with Goto-Kakizaki rats, *Lactobacillus gasseri* SBT2055 increased insulin secretion via increased expression of insulin genes (*Ins1* and *Ins2*) and an increased transcription factor for insulin genes—pancreatic and duodenal homeobox (PDX1)—showing improvement in pancreatic inflammatory status and a decrease in serum glucose levels [168].

Hyperglycaemia is also associated with loss of chloride ions. High intracellular concentrations of chloride ions in pancreatic β-cells are critical for the electrical activity of the β-cell membrane and insulin secretion. The addition of the *Lactobacillus casei* Zhang strain in a rat study increased of chloride channel protein 2 (ClC-2) expression in chloride channels, resulting in reduced chloride ion loss and improved electrical activity of the β-cell membrane and insulin secretion [131].

#### 3.7.3. Kidney

Insulin receptor substrate-1 protein (IRS-1), AKT and endothelial nitric oxide synthase (eNOS) are important components of glucose homeostasis signalling pathways and lipid metabolism in the kidney. In addition, sodium-glucose cotransporter 2 (SGLT2) is the major glucose transporter, while glucose transporter 5 (GLUT5) is the specific fructose transporter [145,169]. In a study on rats fed a high fructose concentration, Korkmaz et al. [170] demonstrated a positive effect of *Lactobacillus plantarum* and *Lactobacillus helveticus* strains on the insulin signalling pathway, inflammatory markers and renal glucose transporters. *Lactobacillus plantarum* and partially *Lactobacillus helveticus* enhanced the expression of IRS -1, AKT and eNOS. In addition, *Lactobacillus plantarum* decreased the level of individual inflammatory factors and the expression of the *IL-6* and *SGLT2* genes, with unchanged expression of the *GLUT5* gene. Various physiological changes caused by the administration of selected probiotics in the pancreas and kidney are shown in Figure 4.

### 3.8. Effect of Probiotics on the Intestine

After ingestion, the intestine is the target organ of the probiotic and therefore of utmost importance for its various mechanisms of action. Figure 5 shows some common effects of probiotics in the intestine and the impact of their action on glucose metabolism and homeostasis.

#### 3.8.1. Effect of Probiotics on the Intestinal Microbiota

A typical intestinal microbiota of an adult consists primarily of a set of six or seven different phyla, of which bacterial species of the phyla Bacteroidetes (Bacteroidota) and Firmicutes (Bacillota) predominate [171]. Bacterial species of the phyla Proteobacteria (Pseudomonadota), Verrucomicrobia (Verrucomicrobiota), Actinobacteria (Actinomycetota) and Euryarchaeota make up a smaller proportion [172]. Dysbiosis is often associated with various pathological conditions, including metabolic diseases such as obesity and type 2 diabetes [13]. In both diseases, the bacteria *Akkermansia muciniphila* and *Faecalibacterium prausnitzii* show a negative correlation [34]. Furthermore, Papadopolous et al. [28] emphasise in their review that dysbiosis of the gut microbiota is the decisive cause for a large number of risk factors for cardiovascular diseases.

Over the last decade, studies in mice have clearly shown a link between the gut microbiota and obesity. The ratio between the number/concentration of the phyla Firmicutes (Bacillota) and Bacteroidetes (Bacteroidota) has shifted in favour of the phylum Firmicutes (Bacillota) in obese mice and mice fed with high-calorie diet [113,173,174]. In addition, bacteria of the phylum Bacteroidetes (Bacteroidota) have fewer genes encoding enzymes relevant for the degradation of lipids and carbohydrates [175]. However, the results of previous studies are very different, even contradictory [123]. They show a positive [176,177,178,179] or negative [180,181] relationship between bacteria of the phyla Firmicutes (Bacillota)/Bacteriodetes (Bacteroidota) and the body mass index (BMI) or statistically non-significant relationships [182,183]. With regard to the higher proportion of *Lactobacillus* bacteria and the lower proportion of *Bifidobacterium* bacteria in obese people, the literature is consistent [184]. In addition, a decrease in metabolic values as well as weight gain after taking probiotics was observed in obese adults [123].

Although type 2 diabetes can be considered a pathological condition associated with obesity, metagenomic studies reveal a typical structure of the intestinal microbiota in patients with type 2 diabetes [24,34]. In addition, Larsen et al. [185] have demonstrated a positive correlation between the phyla Bacteroidetes (Bacteroidota)/Firmicutes (Bacillota) ratio and serum glucose levels. However, it has been shown that the microbiota of these individuals often contains a lower proportion of butyric acid producing bacteria and thus several opportunistic bacteria. Furthermore, by analysing the structure of the gut microbiota of patients with type 2 diabetes or those prone to developing the disease, physicians can improve the prognosis or course of the disease by preventing a deterioration in glycaemic status, and thus, the development of insulin resistance [13]. However, it is recommended to include functional foods with probiotic properties in the daily diet of patients with type 2 diabetes to improve metabolic imbalance and control and prevent the severity of the disease [27,123].

In vivo human studies

In a human study, Ivey et al. [186] showed no effect of consuming probiotic yoghurt and probiotic capsules on the composition of the microbiota. They reasoned that the performance and efficacy of probiotic bacteria depend on the specific conditions in the gut. Indeed, the expression of probiotic genes is influenced by both the genotype of the host and the genotype of the interacting gut bacteria. In addition, the host’s diet can also influence the metabolism of probiotics [186]. In another human study, no change in the composition of the gut microbiota was observed after probiotic addition, which was explained by the fact that the probiotic metabolic response depends on the initial composition of the host gut microbiota, i.e., before the start of therapy [54]. An eight-week randomised control trial in children and adolescents with obesity showed a change in microbiota (reduced number of *Escherichia coli*), weight and an improvement in insulin sensitivity and certain metabolic parameters after supplementation with *Bifidobacterium breve* BR03 and B632 [187]. In patients with type 2 diabetes, the use of different *Lactobacillus reuteri* ADR-1/3 strains affected the modulation of the intestinal microbiota (increase in fecal *Lactobacillus reuteri* levels), resulting in HbA1c reduction and alleviation of type 2 diabetes symptoms [51].

Early pregnancy is generally characterised by glucose and insulin levels being within the normal range [188]. However, in some cases, an increase in serum insulin levels is observed in late pregnancy, often associated with insulin resistance, which can lead to glucose hypersensitivity and type 2 diabetes—gestational diabetes mellitus (GDM) [189]. Insulin resistance is thus the cause of inflammation (increase in various inflammatory factors) and depends on the structure of the intestinal microbiota, which has a significant influence on insulin metabolism. GDM not only has negative effects on the mother, but also on the newborn, who shows signs of macrosomia and hypoglycaemia [190]. Systematic studies [190] on the use of probiotics (mostly *Lactobacillus* and *Bifidobacterium*) indicate their beneficial effect on GDM in the period between pregnancies. Probiotics can alter control over the gut microbiota [189], helping to modulate the immune system, reduce inflammatory factors and improve glucose and blood sugar control [189,191,192,193]. Ebrahimi et al. [194] also showed a positive effect of probiotic yoghurt consumption on blood glucose control, i.e., a significant decrease in blood glucose and HbA1c levels in pregnant women. On the other hand, Shahriari et al. [195] showed neither a significant reduction in the risk of GDM nor an improvement in other neonatal and maternal outcomes after oral administration of a probiotic supplement mixture.

The exact effects and thus the relationship between probiotics and the intestinal microbiota are still not fully understood, indicating some conflicting studies. It seems that the relationship between the intestinal microbiota and probiotics is unique to each individual and should be studied as such [196].

In vivo animal model studies

In the studies listed below, changes in the Bacteroidetes (Bacteroidota)/Firmicutes (Bacillota) ratio and in the relative abundance of the two phyla after different probiotic administrations may represent the best parameter for evaluating the beneficial effects of probiotics on the gut microbiota. Restoration of the two phyla was reported regardless of the pathological conditions of the animal. Since it is already known that there is a positive correlation between the ratio/relative abundance of the two phyla and glucose in some pathological conditions [192], it is not surprising that the administration of certain probiotics could improve or restore glucose tolerance and insulin sensitivity, possibly through direct or indirect modulation of the host gut microbiota. This includes increasing the relative abundance of Bacteroidetes (Bacteroidota) and decreasing that of Firmicutes (Bacillota), resulting in a decrease in the Firmicutes (Bacillota)/Bacteroidetes (Bacteroidota) ratio. Table 3 summarises some modulatory effects of selected probiotics administration on the microbiota in relation to glucose metabolism and homeostasis in animal model studies.

Zhang et al. [91] showed a positive correlation between phyla Bacteroidetes (Bacteroidota)/Firmicutes (Bacillota) ratio and serum glucose levels after administration of *Lactobacillus casei* LC89 to diabetic mice. They even suggested that the reduction in Firmicutes (Bacillota) and increase in Bacteroidetes (Bacteroidota) could be an additional factor in improving glucose tolerance. After 11 weeks of treatment with *Lactobacillus plantarum* HT121 [197] and Q180 [146], a reduction in body weight and an improvement in glucose tolerance were observed in mice fed with a high-fat diet, which could correlate with the restored Firmicutes (Bacillota)/Bacteroidetes (especially *Akkermansia*, *Lactobacillus* and *Ruminococcus* [197]) ratio. Li et al. [198] showed in rats that administration of the probiotic *Lactobacillus casei* CCFM419 increased the proportion of bacteria belonging to genus *Allobaculum* and *Bacteroides*. These changes may help to alleviate type 2 diabetes. In another study in diabetic mice, changes in the intestinal microbiota were observed after treatment with *Lactobacillus plantarum* HAC01, i.e., an increase in the Verrucomicrobia (Verrucomicrobiota) phylum (*Akkermansia muciniphila*) and a decrease in the Proteobacteria (Pseudomonadota) phylum (*Dusulfovibrionaceae* family). The increased levels of *Akkermansia* may have contributed to increasing insulin sensitivity [103]. Furthermore, in a study with diabetic mice [101], 14 compound probiotics reduced the number of harmful bacteria such as *Bacteroides thetaiotaomicron*, whose polysaccharose degradation leads to enhanced glucose and energy absorption, resulting in hyperglycaemia. Li et al. [199] have demonstrated an increase in acetate- and butyrate-producing bacteria such as *Clostridium leptum*, *Bacteroides* and *Prevotella*, which may be related to improved blood glucose and glucose intolerance. Five out of nine different probiotic strains tested in mice with high-fat diet and streptozoticin-induced type 2 diabetes increased short-chain fatty acids (SCFA)-producing gut bacteria such as *Bacteroidales* S24-7, *Parabacteroides*, *Mucispirillum* and *Coprococcus*. In addition, these strains have been directly associated with improving glucose metabolism and alleviating insulin resistance [77]. A correlation between plasma glucose and the Firmicutes (Bacillota)/Bacteroidetes (Bacteroidota) ratio has been proposed also by Yan et al. They showed that the administration of two strains of *Lactobacillus acidophilus* increased the relative abundance of the genera *Blautia*, *Roseburia* and *Anaerotruncus*, while Gram-negative bacteria of the genera *Desulfovibrio*, *Alistipes* and *Bacteroides* decreased [110]. Administration of heated-killed *Streptococcus thermophilus* improved some glycaemic parameters of diabetic rats, while increasing the levels of some beneficial bacteria of the genera *Ruminococcaceae*, *Veillonella*, *Coprococcus* and *Barnesiella* [84]. A reduction in blood glucose levels due to the enrichment of the genus *Alistipes* following the administration of engineered *Lactobacillus plantarum*-pMG36e-GLP-1 in diabetic monkeys has been suggested. Moreover, the probiotic reduced the phylum Bacteroidetes (Bacteroidota), particularly the bacterial species of *Prevotella*, whose association with type 2 diabetes has already been confirmed [200].

Zheng et al. [201] showed that probiotic supplementation of *Lactobacillus rhamnosus* LGG and *Bifidobacterium animalis* subsp. *lactis* Bb12 alleviated GDM in pregnant rats and showed an improvement in fasting blood glucose levels due to restoration of the gut microbiota. In fact, the abundance of Firmicutes (Bacillota) and Actinobacteria (Actinomycetota) increased. Furthermore, the probiotic mechanism related to GDM might be linked to the inhibition of carbohydrate metabolism and membrane transport pathways.

#### 3.8.2. Effect of Probiotics on the Formation of SCFA

SCFA are non-digestive (human) metabolic nutrients derived mainly from the fermentation of resistant carbohydrates. In humans, SCFA account for 10% of daily energy expenditure and are predominantly of three types, acetic, propionic and butyric acids [202]. SCFA have been proposed as an important link between health and disease, particularly as energy carriers, modulators of cell proliferation and differentiation, and as immune modulators [203] and therefore as an important research component in analytical methods [204].

Yadav et al. [205] have shown in mouse models that the addition of the probiotic VSL#3 has a positive effect on obesity and diabetes. The probiotic reduced appetite, body weight and insulin resistance and increased glucose tolerance by modulating the composition of the gut microbiota and increased butyrate levels. Wang et al. [155] also hypothesised that the use of the probiotic Lactobacillus casei CCF419 influenced the regulation of the structure of the gut microbiota and thus the composition of SCFA in favour of increased acetate and butyrate. A very similar effect was observed in a study with high-fat and streptozotocin-induced diabetic mice. After administration of Lactobacillus plantarum HAC01, a restoration of the gut microbiota and an increase in butyric acid levels were observed [103]. Five different probiotic strains out of nine tested increased SCFA levels in the faeces of mice with high-fat diets and streptozoticin-induced type 2 diabetes, while showing a hypoglycaemic effect, reducing inflammation and decreasing insulin resistance. An increase in SCFA levels was associated with increased abundance in the SCFA-producing gut microbiota [77].

One of the potential mechanisms of SCFA in the prevention of type 2 diabetes involves intestinal gluconeogenesis, which is directly linked to the periportal nervous system via a specific signalling system. Thus, intestinal gluconeogenesis is inversely correlated with the risk factor for type 2 diabetes, as shown by several studies demonstrating a decrease in body weight, secretion of glucose in the liver and improved glycaemic control [24]. Furthermore, butyrate directly enhances the expression of genes involved in intestinal gluconeogenesis via free fatty acid receptors (FFAR). On the other hand, propionic acid (propionate) binds to FFAR3 in the portal vein nerves. Both signalling pathways are important for the glucose portal sensing process [206]. Toejing et al. [82] and Yan et al. [110] have looked at the importance of increased SCFA levels in the caecum of rats and mice for glucose metabolism after administration of *Lactobacillus paracasei* HII01 and *Lactobacillus acidophilus*, respectively.

GPR43 is one of the specific G protein-coupled receptors (GPRs) and it is known that its activation by acetate and propionate is more sensitive than by butyrate [207]. In a study in mice, Horiuchi et al. suggested that increased levels of SCFA (especially acetate) by administration of *Bifidobacterium animalis* subsp. *lactis* GCL2505 improved host glucose tolerance and energy expenditure and overcame body fat accumulation by activating GPR43 [208]. One of the several antidiabetic effects demonstrated in diabetic mice after administration of 14 compound probiotics was improved insulin secretion via glucose-triggered secretion of glucagon-like polypeptide (GLP)-1 and upregulation of GPR43/41 [100]. Lu et al. [209] suggested that a composite of three probiotics could act on tumour suppression in colorectal cancer cells by inhibiting tumourigenesis and glucose metabolism through GPR43 activation. However, the activation may have resulted in decreased expression of some glycolytic genes, which in turn resulted in decreased glucose metabolism.

#### 3.8.3. Effect of Probiotics on the Secretion of Intestinal Incretins

Incretins are intestinal hormones secreted by enteroendocrine cells. The best-known representatives, GLP-1 and the glucose-dependent insulinotropic polypeptide, i.e., gastric inhibitory polypeptide (GIP), are involved in the physiological regulation of glucose homeostasis. When glucose increases, (e.g., due to food intake), the two hormones are released into the blood to increase the synthesis and release of insulin from pancreatic cells [210]. In addition, GLP-1 reduces the secretion of the hormone glucagon from the α-cells of the pancreas, which helps to control blood glucose levels. In addition, GLP-1 has an effect on decreased appetite, proliferation of pancreatic β-cells, gastrointestinal motility and gastric emptying. However, the endogenous enzyme dipeptidyl peptidase-4 (DPP-4) protease ensures rapid degradation of incretins in the body, which consequently affects their short-term activity in the blood. However, secretion of incretins can be significantly impaired in conditions such as obesity and type 2 diabetes [211].

The importance of GLP-1 in relation to blood glucose levels was demonstrated in a study on diabetic monkeys. Luo et al. showed the blood sugar-lowering effect after administration of a genetically modified strain *Lactobacillus plantarum*-pMG36e-GLP-1, which constantly expresses GLP-1 [200].

In a clinical trial with 21 subjects, it was shown that a 4-week intake of *Lactobacillus reuteri* SD586521 increased the secretion of GLP-1, which was associated with an increase in insulin and C-peptide concentrations. Therefore, the authors presented the probiotic *Lactobacillus reuteri* SD586521 as a new therapeutic approach to improve glucose-dependent insulin secretion [212].

Several studies in animals and in vitro models have demonstrated the effect of probiotics (of the species *Lactobacillus*) on increased incretin secretion (GLP-1) and consequently glucose control, as well as reducing signs of obesity and/or type 2 diabetes, i.e., postprandial glucose and insulin resistance [100,101,147,155,213,214].

In a study in rats, Yamano et al. [215] presented the potential mechanism of action of the probiotic *Lactobacillus johnsonii* La1. The probiotic appears to inhibit the activity of the adrenal sympathetic nervous system. They suggested that the lowering of blood glucose and glucagon concentrations was due to the probiotic altering the activity of the autonomic nervous system.

PDX-1, an insulin promoter (stimulating) factor 1, is involved in the glucose-dependent regulation of insulin gene transcription and is essential for the development of pancreatic exocrine and endocrine cells, including β-cells [216]. In an in vitro study, Duan et al. [217] demonstrated an insulinotropic effect of *Escherichia coli* Nissle 1917 on the secretion of GLP-1 and transcription activator PDX-1. The bacterial supernatant has been shown to stimulate intestinal epithelial cells to secrete insulin. Moreover, they also emphasised the importance of optimised conditions to achieve secretion of insulin levels, which is crucial for normal metabolism.

In an in vitro study, Belguesmia et al. [218] assumed that some strains of the genus Lactobacillus are able to degrade certain gut hormones through serine peptidase action. In this way, the probiotic can indirectly influence glucose metabolism and homeostasis.

#### 3.8.4. Effect of Probiotics on Permeability and Integrity of the Intestinal Wall

Lim et al. [132] demonstrated in a mouse model that *Lactobacillus sakei* OK67 alleviated hyperglycaemia induced by a high-fat diet by reducing inflammation and increasing the expression of tight junction (TJ) proteins in the intestinal wall. Because probiotics affect the composition of the gut microflora, studies in rats [219,220] and mice [100,101,105,107,110] have shown that they improve the integrity of the intestinal epithelium and enhance the local immune response. In fact, the immunomodulatory effect of various probiotics led to a reduction in blood glucose levels. Moreover, in studies in rats [219,220] and a study in mice [105], probiotics inhibited intestinal toll-like receptor 4 (TLR 4) signalling [105,219,220] and activated toll-like receptor 2 (TLR 2) [105], which consequently had an effect on reducing some inflammatory processes, increasing insulin resistance and attenuating impaired glucose tolerance. In addition, decreased TLR 4 and increased TLR 2 mRNA levels were associated with decreased mRNA levels of the pro-inflammatory cytokines TNF-α and IL-6 [105].

An in vitro study with different *Lactobacillus* strains has shown that *Lactobacillus gasseri* ICVB396 contributes at most to the restoration and strengthening of the epithelial wall of the intestine in the Caco2 cell line [218].

TNF-α and IL-6 were investigated in an in vitro study of LPS-induced inflammation in Caco-2 cells as well as in the intestinal macrophage line RAW264.7 after addition of three different probiotic strains. A reduction in both pro-inflammatory parameters proved an improvement in intestinal restoration and insulin resistance [147].

#### 3.8.5. Effect of Probiotics on Glucose Transport in Intestine

Rooj et al. [221] have demonstrated the interaction of *Lactobacillus acidophilus* and other species of the genus *Lactobacillus* with the intestinal epithelium in an in vitro study. After the short-term exposure of intestinal cells to a bacterial supernatant, reduced glucose accumulation was observed.

In CaCo-2 cells, Hsieh et al. [93] showed that Lactobacillus salivarus AP-32 and Lactobacillus reuteri GL-104 downregulated two major hexose transporters in the brush border gut membrane–sodium–glucose cotransporter 1 (SGLT1) and GLUT5. Consequently, both transporters automatically blocked sugar uptake in the gut. However, the expression of the basolateral glucose transporter GLUT2 was upregulated, leading to speculation that the upregulation of GLUT2 may have occurred in response to the reduction in SGLT1 and GLUT5 expression.

Another in vitro study on Caco-2 cells showed possible effects of three different strains on enhanced glucose uptake, possibly through upregulation of the PI3K/AKT pathway [147].

In our recent in vitro study [222] we tested the hypothesis of the effects of selected *Lactobacillus* probiotics on the regulation of the enterocytic transporters SGLT1 and GLUT2 in two different intestinal epithelial models. We demonstrated that the changes in expression of both transporters were strain- and cell-line-specific. Moreover, *Lactobacillus plantarum* PCS26 and *Lactobacillus acidophilus* downregulated GLUT2 expression, which was also observed by decreased transepithelial glucose transport rates, leading to reduction in glucose absorption.

Wang et al. showed that administration of eight probiotic strains decreased the expression of SGLT1 and GLUT2 in STZ-induced diabetic mice, indicating their effect on reducing digestion and glucose absorption in the gut [96].

## 4. Concluding Remarks and Future Perspectives

Our review shows that probiotics play an important role in health and diseases related to glucose metabolism. Numerous in vivo studies in humans and animals, as well as studies in vitro, have clearly shown that probiotics can exert their direct or indirect effects on glucose and its metabolism in almost all tissues and organs of the body. Moreover, their interaction with the physiological functions of various brain–hormone axes has proven to play an important role and act in several areas of the species’ body. Although most of the proposed probiotic effects have been shown to be beneficial and positive results have been obtained, there still seems to be a large amount of variation in results, especially between human studies, due to a variety of aspects on which one may or may not have specific effects.

A standardisation of methodological approaches in in vivo studies can definitely contribute to a better understanding of the potential probiotic mechanisms and the consequences of their effects. Therefore, the use of one or more strains (of the same), the administration doses (amount) and their duration, the number of patients/animals, their age, their diet and their possible physical activity (diversity within the study) should be considered more seriously before therapy. Furthermore, many methodological approaches still lack possible probiotic side effects and additional testing of other metabolic factors affecting glycaemic control. Given the above limitations, research into probiotic mechanisms and their potentially broader role in body functioning could be improved. Nevertheless, numerous effects of probiotics on glucose metabolism have been studied in various pathological metabolic conditions such as type 2 diabetes, metabolic syndrome, insulin resistance, glucose intolerance and hyperglycaemia. In this regard, since the initiation of any probiotic therapy, a disease state, its severity or potential risk factors associated with its outcome should be of enormous importance. In vivo animal models should consider whether induced or naturally occurring diseases were included in the study, as they may have different effects on probiotics and thus on their effects in animals. However, the animal studies appeared to be more homogeneous with regard to the effects of probiotics on glucose metabolism and homeostasis.

Nevertheless, there are some limitations in in vivo human studies that may always compromise the variety of probiotic effects, regardless of their potentially unique positive outcomes, especially in glucose metabolism and its homeostasis. Humans are to some extent genetically different and not alike due to environmental and geographical differences. They also have their own dietary habits that greatly affect their microbiota and probiotics as well. Therefore, more consideration should be given to the status of the microbiota, which is considered as unique as a fingerprint, before probiotic therapy. Last but not least, specific alterations of the microbiota have been associated with various pathological diseases affecting glucose metabolism and its homeostasis, and their modulation through probiotic administration could be important in studies on disease-targeted therapies. However, the number of human studies on this topic is still small, just as the studies on the exact effects (short- or long-term) of probiotics on the intestinal microbiota that influence glucose homeostasis are not fully presented and clarified. The fact that modification in intestinal microbiota after the administration of probiotics varies depending on the type of model (and its pathological/physiological conditions) and the experimental conditions is also something we have learned from the studies on animal models. Although more interactions between probiotics–intestinal microbiota–glucose metabolism and its homeostasis through the microbiota-related intestinal wall, specific microbiota-related SCFAs affecting hormone and SCFA secretion and glucose/lipid metabolism-related signalling routes are proposed, the exact mechanisms of how probiotics work are still being thoroughly investigated.

It has already been proven that probiotics have species- and strain-specific effects. In terms of effects on glucose metabolism and its homeostasis, studies with the genus *Lactobacillus* still predominate, followed by *Bifidobacterium*. However, studies on novel probiotic strains belonging to the genera of *Akkermansia*, *Faecalibacterium*, *Prevotella* have limited publications, but they have shown potential efficacy and should therefore be the subject of further future studies.

Moreover, further studies on the interactions between probiotics and the host are needed. Indeed, for proper interactions to occur, the involvement of commensals appears to be essential. Probiotics are microorganisms that serve to achieve a balanced co-existence and simultaneous synergistic effect with the host microbiota. The relationship between the two, as well as their effects on the host immune system, are therefore unique and should be studied as such when it comes to new research on the probiotics mechanisms of action.

Since energy metabolism is related to gluconeogenesis and the production of ketone bodies during starvation or diets with very low availability of digestible carbohydrates [223], more importance of analysing ketone bodies in the context of hypo- and more recently hyperglycaemic states in in vivo studies should be considered. Furthermore, the effect of probiotics on ketone body levels and energy metabolism reduction as well as their positive correlation, has been demonstrated in animal models [224].

Although the use of probiotics and prebiotics as modulators of the intestinal microbiota on the host still has the most successful short- and long-term effects, especially in glucose-related diseases, faecal microbiota transplantation has raised many promising expectations in this field [225,226]. Short-term improvement in some parameters of glucose and lipid metabolism has been reported [227,228,229], but the long-term effects in terms of their side effects still need to be carefully studied, as it seems that more than one treatment should be considered to achieve adequate results.

## Figures and Tables

**Figure 1 life-12-01187-f001:**
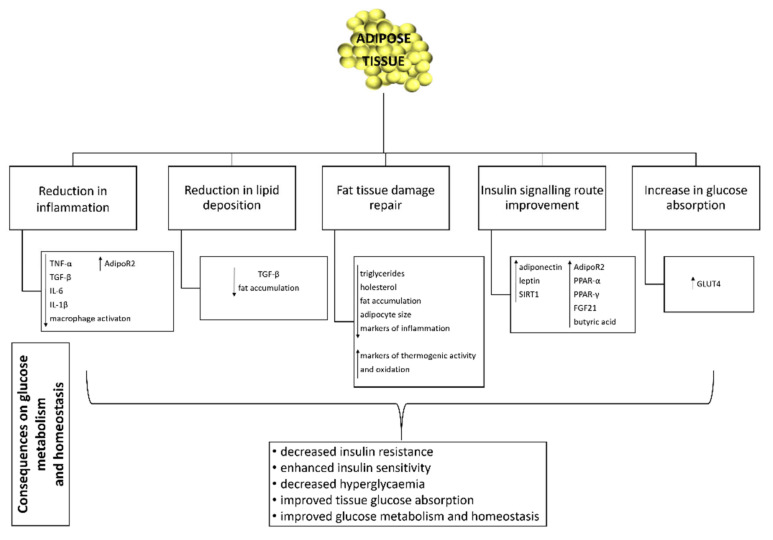
Effect of probiotics on adipose tissue. TNF-α: tumour necrosis factor alpha; TGF-β: transforming growth factor beta; IL-6: interleukin-6; IL-1β: interleukin 1 beta; AdipoR2; adiponectin receptor 2; TGF-β: transforming growth factor beta; SIRT1: sirtuin (silent mating type information regulation 2 homolog) 1; PPARα: peroxisome proliferator-activated receptor α; PPAR-γ: peroxisome proliferator-activated receptor gamma; FGF21: fibroblast growth factor 21; GLUT4: glucose transporter 4; ↓: decrease; ↑: increase.

**Figure 2 life-12-01187-f002:**
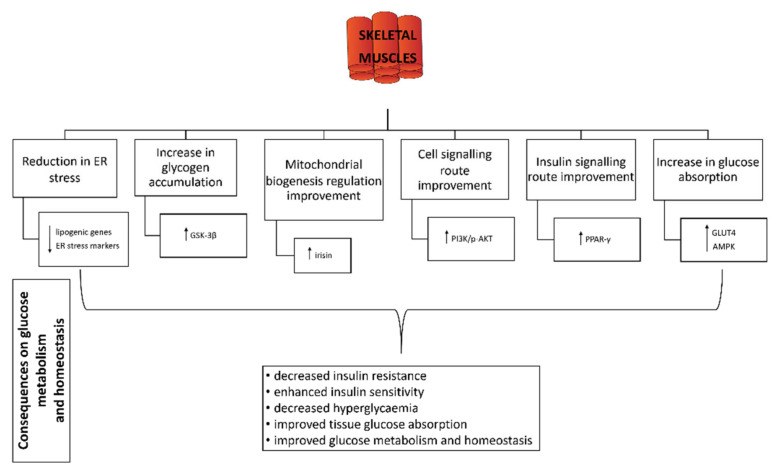
Effect of probiotics on skeletal muscles. GSK-3β: glycogen synthase kinase 3 beta; PI3K/AKT: phosphoinositide 3-kinase/phosphorylated protein kinase B signalling pathway; PPAR-γ: peroxisome proliferator-activated receptor gamma; GLUT4: glucose transporter 4; AMPK: 5′ adenosine monophosphate-activated protein kinase; ↓: decrease; ↑: increase.

**Figure 3 life-12-01187-f003:**
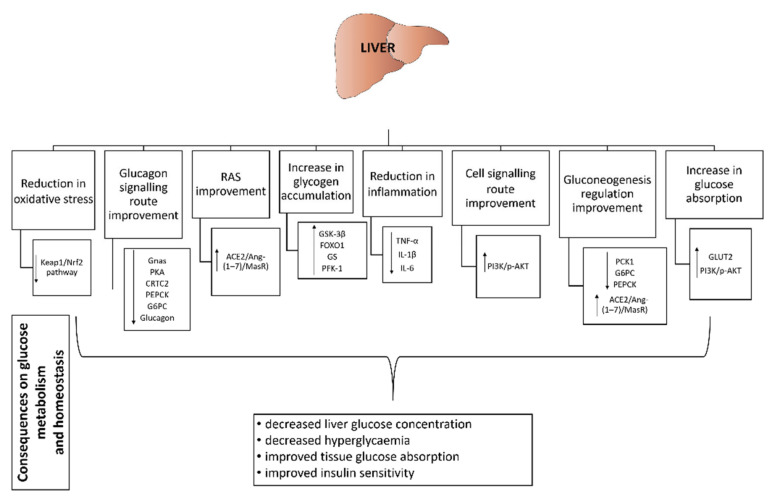
Effect of probiotics on liver. Keap1/Nrf2: Kelch-like ECH-associated protein 1/Nuclear factor-erythroid factor 2-related factor 2; Gnas: Gs protein alpha-subunit; PKA: cAmp-dependent protein kinase; CRTC2: CREB-regulated transcriptional coactivator 2; PEPCK: phosphoenolpyruvate carboxykinase; G6PC: glucose-6-phosphatase catalytic subunit; ACE2/Ang-(1–7)/MasR): angiotensin-converting enzyme 2/angiotensin-(1–7)/mitochondrial assembly receptor; GSK-3β: glycogen synthase kinase 3 beta; FOXO1: Forehead Box O1; GS: glycogen synthase; PFK-1: phosphofructokinase-1; TNF-α: tumour necrosis factor alpha; IL-6: interleukin-6; IL-1β: interleukin 1 beta; PI3K/AKT: phosphoinositide 3-kinase/phosphorylated protein kinase B signalling pathway; PCK1: phosphoenolpyruvate carboxykinase 1; GLUT2: glucose transporter 2; ↓: decrease; ↑: increase.

**Figure 4 life-12-01187-f004:**
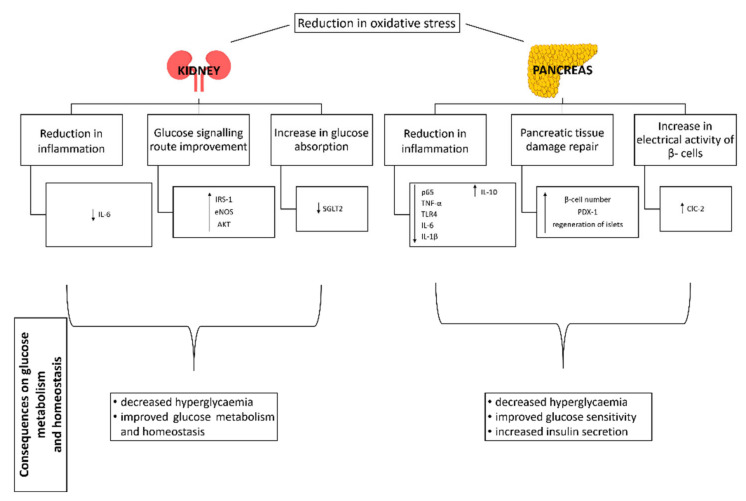
Effect of probiotics on kidney and pancreas. IL-6: interleukin-6; IRS-1: insulin receptor substrate 1 protein; eNOS: endothelial nitric oxide synthase; AKT: protein kinase B; SGLT2: sodium-glucose cotransporter 2; p65: transcription nuclear factor NF-kappa-B p65 subunit; TNF-α: tumour necrosis factor alpha; TLR4: toll-like receptor 4; IL-1β: interleukin 1 beta; IL-10: interleukin 10; PDX-1: insulin promoter factor 1; ClC-2: chloride channel protein 2; ↓: decrease; ↑: increase.

**Figure 5 life-12-01187-f005:**
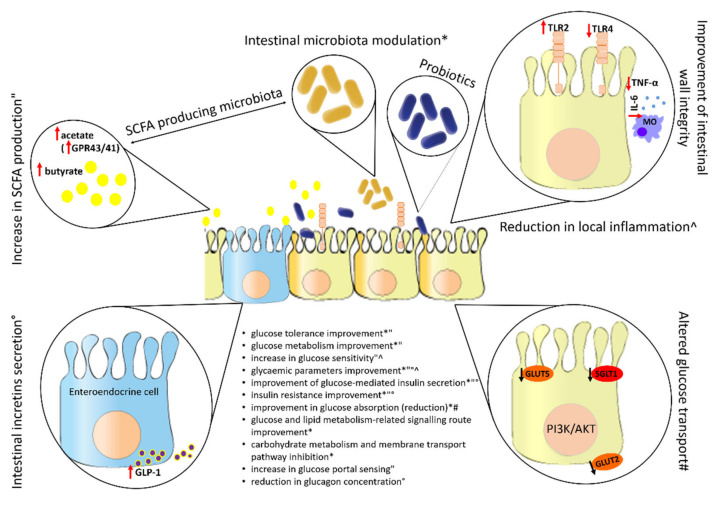
Effect of probiotics on intestine. SCFA: short-chain fatty acids; PI3K/AKT: phosphoinositide 3-kinase/phosphorylated protein kinase B signalling pathway; GPR43/41: G protein-coupled receptor 43/41; GLP-1: glucagon-like polypeptide-1; TLR4: toll-like receptor 4; TLR2: toll-like receptor 2; TNF-α: tumour necrosis factor alpha; IL-6: interleukin-6; GLUT2: glucose transporter 2; GLUT5: glucose transporter 5; SGLT1: sodium-glucose cotransporter 1; ↓: decrease; ↑: increase; *, ", ^, °, #: each symbol stands for a different probiotic effect, which has a corresponding effect on glucose metabolism and homeostasis.

**Table 1 life-12-01187-t001:** Positive probiotic effects on some blood parameters in selected in vivo human studies.

Altered Blood Parameters	Type of Probiotic	Physiological/ Pathological Condition	Reference
↓ Postprandial blood glucose	*Lacticaseibacillus casei* 01 (paraprobiotic)	None	[42]
↓ FPG	*Lactobacillus acidophilus* La-5 and *Bifidobacterium animalis* subsp *lactis* BB-12	T2D	[43]
	*Lactobacillus acidophilus La-5* and *Bifidobacterium animalis subsp* *lactis* BB-12	Metabolic syndrome	[44]
	*Bifidobacterium longum* APC1472	Healthy overweight/obese	[45]
	*Lactobacillus plantarum* OLL2712	Pre-diabetic	[46]
	Ecologic^®^ Barrier (multi-strain probiotics)	T2D	[47]
	*Lactobacillus casei* 01	T2D	[48]
	*Lactobacillus casei*	T2D	[49]
↓ HbA1c	*Lactobacillus acidophilus* La-5 and *Bifidobacterium animalis* subsp *lactis* BB-12	T2D	[50]
	*Lactobacillus reuteri* ADR-1	T2D	[51]
	“Symbiter” (multi-strain)	T2D	[52]
	*Lactobacillus acidophilus* La-5 and *Bifidobacterium animalis* subsp *lactis* BB-12	T2D	[43]
	*Lactobacillus acidophilus* La-5 and *Bifidobacterium animalis* subsp *lactis* BB-12	T2D	[53]
HOMA-IR improvement (reduction)	*Lactobacillus plantarum* OLL2712	Pre-diabetic	[46]
	“Symbiter” (multi-strain)	T2D	[52]
	Ecologic^®^ Barrier (multi-strain probiotics)	T2D	[47]
	*Lactobacillus casei* 01	T2D	[48]
	*Lactobacillus casei*	T2D	[49]
	*Lactobacillus acidophilus* La-5 and *Bifidobacterium animalis* subsp *lactis* BB-12	Metabolic syndrome	[44]
↓ Fructosamine	*Lactobacillus acidophilus* La-5 and *Bifidobacterium animalis* subsp *lactis* BB-12	T2D	[50]
↓ Fasting insulin levels	Ecologic^®^ Barrier (multi-strain probiotics)	T2D	[47]
	*Lactobacillus casei*	T2D	[49]
	*Lactobacillus casei* 01	T2D	[48]
	*Lactobacillus acidophilus* La-5 and *Bifidobacterium animalis* subsp *lactis* BB-12	Metabolic syndrome	[44]
Insulin sensitivity improvement (↑ ISI, QUICKI test or others)	*Lactobacillus reuteri* DSM 17938	T2D	[54]
	Ecologic^®^ Barrier (multi-strain probiotics)	T2D	[47]
	*Lactobacillus acidophilus* NCFM	T2D	[55]
	*Lactobacillus plantarum* OLL2712	Pre-diabetic	[46]
	*Lactobacillus acidophilus* La-5 and *Bifidobacterium animalis* subsp *lactis* BB-12	Metabolic syndrome	[44]
↓ Ghrelin	*Bifidobacterium longum* APC1472	Healthy overweight/obese	[45]
↓ Cortisol	*Bifidobacterium longum* APC1472	Healthy overweight/obese	[45]
↓ Glycoalbumin	*Lactobacillus plantarum* OLL2712	Pre-diabetic	[46]
↓ Fetuin-A	*Lactobacillus casei* 01	T2D	[48]
	*Lactobacillus casei*	T2D	[49]
↑ SIRTs	*Lactobacillus casei* 01	T2D	[48]
	*Lactobacillus casei*	T2D	[49]
↓ IL-6	*Lactobacillus plantarum* OLL2712	Pre-diabetic	[46]
	Ecologic^®^ Barrier (multi-strain probiotics)	T2D	[47]
↓ TNF-α	“Symbiter” (multi-strain)	T2D	[52]
	*Lactobacillus plantarum* OLL2712	Pre-diabetic	[46]
	*Lactobacillus acidophilus* La-5 and *Bifidobacterium animalis* subsp *lactis* BB-12	T2D	[50]
	Ecologic^®^ Barrier (multi-strain probiotics)	T2D	[47]
↓ IL-1β	“Symbiter” (multi-strain)	T2D	[52]
	*Lactobacillus plantarum* OLL2712	Pre-diabetic	[46]
	*Lactobacillus reuteri* ADR-3	T2D	[51]
↓ MCP-1	*Lactobacillus plantarum* OLL2712	Pre-diabetic	[46]

T2D: type 2 diabetes; FPG: fasting plasma glucose; HbA1c: glycohemoglobin; HOMA-IR: homeostasis model assessment-estimated insulin resistance; ISI: insulin sensitivity index; HFD: high-fat diet; QUICKI: quantitative insulin sensitivity check index; OGTT: Oral Glucose Tolerance Test; SIRTs: sirtuins; IL-6: interleukin-6; IL-1β: interleukin 1 beta; TNF-α: tumour necrosis factor alpha; MCP-1: monocyte chemotactic protein-1; ↓: decrease; ↑: increase.

**Table 2 life-12-01187-t002:** Positive probiotic effects on some blood parameters in selected animal models studies.

Altered Blood Parameters	Type of Probiotic	Species	Physiological/Pathological Condition and/or Diet	Reference
↓ Postprandial blood glucose	*Lactobacillus rhamnosus* (YC, 7-1), *Bifidobacterium adolescentis* (N3, 7-2) and *Bifidobacterium bifidum* (M2)	Mouse	HFD STZ-induced diabetes	[77]
	*Lactobacillus plantarum* CCFM0236	Mouse	STZ-induced diabetes	[78]
↓ FPG	*Lactobacillus casei* and *Bifidobacterium bifidum* (alone or combination of both)	rat	STZ-induced diabetes	[79]
	*Lactobacillus plantarum* TN627	Rat	Alloxan-induced diabetes	[80]
	*Bifidobacterium animalis* 01	Rat	High-fat chow diet STZ-induced diabetes	[81]
	*Lactobacillus paracasei* HII01	Rat	HFD STZ-induced diabetes	[82]
	*Lactobacillus rhamnosus* (YC, 7-1), *Bifidobacterium adolescentis* (N3) and *Bifidobacterium bifidum* (M2)	Mouse	HFD STZ-induced diabetes	[77]
	*Lactobacillus rhamnosus* (BSL and R23)	Rat	STZ-induced diabetes	[83]
	Heat-inactivated *Streptococcus* *thermophilus*	Rat	HFD ZDF diabetes	[84]
	*Lactiplantibacillus plantarum* IMC 510	Rat	Diet-induced obesity	[85]
	*Lactobacillus fermentum* RS-2	Rat	Alloxan-induced diabetes	[86]
	*Bacillus licheniformis* Zhengchangsheng^®^	Mouse	HFD-induced obesity	[87]
	*Weissella cibaria* MG5285, *Limosilactobacillus reuteri* MG5149, *Lacticaseibacillus rhamnosus* MG4502, *Lactobacillus gasseri* MG4524	Mouse	HFD-induced obesity	[88]
	Probioglu^TM^	Rat	STZ-induced diabetes	[89]
	*Bacillus toyonensis* SAU-19	Mouse	HFD ZDF-induced diabetes	[90]
	*Lactobacillus casei* LC89	Mouse	STZ-induced diabetes	[91]
	Lactibiane Tolérance^®^	Mouse	HFD	[92]
	*Lactobacillus salivarus* AP-32 *Lactobacillus reuteri* GL-104	Mouse	Db/db obesity	[93]
	*Lactobacillus plantarum* CCFM0236	Mouse	STZ-induced diabetes	[78]
	*Lactobacillus plantarum* YJ7	Mouse	HFD	[94]
	*Bifidobacterium longum* BR-108 (inactivated)	Mouse	Obese diabetes	[95]
	*L. rhamnosus* L12, *L. acidophilus*, *L. plantarum* HM218749, *B. animalis* subsp. *lactis* LPL-RH, *B. longum* subsp. *longum* BAMA-B05/BAu-B1024)	Mouse	Glucose STZ-induced diabetes	[96]
	*L. rhamnosus* L12, *L. acidophilus*, *L. plantarum* HM218749, *B. animalis* subsp. *lactis* LPL-RH, *B. longum* subsp. *longum* BAMA-B05/BAu-B1024)	Mouse	Glucose water -induced diabetes	[96]
	*Lactobacillus plantarum* DSM 15313	Mouse	HFD-induced obesity	[97]
	*Lactobacillus plantarum* OLL2712	Mouse	HFD-induced obesity	[98]
	*Lactobacillus casei* Shirota YIT 9029	Mouse	HFD-induced obesity	[99]
	14 composite probiotics	Mouse	Db/db diabetes	[100]
	14 composite probiotics	Mouse	Db/db diabetes	[101]
	*Bifidobacterium longum*	Mouse	HFD-induced obesity	[102]
	*Lactobacillus plantarum* HAC01	Mouse	HFD STZ-induced diabetes	[103]
	*Lactobacillus paracasei* NL41	Rat	HFD STZ-induced diabetes	[104]
	*Akkermansia municiphila* and its extracellular vesicles	Mouse	HFD	[105]
	VSL#3 composite probiotics	Mouse	HFD-induced obesity	[106]
	*Lactobacillus paracasei* DTA81	Mouse	HFD	[107]
	*Lactobacillus acidophilus* SJLH001	Mouse	HFD	[108]
	*Lactobacillus casei* Zhang	Rat	Hyperinsulinemia	[109]
	*Lactobacillus acidophilus* KLDS1.1003 and KLDS1.0901	Mouse	HFD STZ-induced diabetes	[110]
↓ HbA1c	*Lactobacillus casei* and *Bifidobacterium bifidum* (alone or combination of both)	Rat	STZ-induced diabetes	[79]
	*Lactobacillus reuteri* ADR-1	Rat	High-fructose diet	[51]
	*Bifidobacterium animalis* 01	Rat	High-fat chow diet STZ-induced diabetes	[81]
	Heat-inactivated *Streptococcus* *thermophilus*	Rat	HFD ZDF diabetes	[84]
	*Lactobacillus plantarum* CCFM0236	Mouse	STZ-induced diabetes	[78]
	14 composite probiotics	Mouse	Db/db diabetes	[100]
	14 composite probiotics	Mouse	Db/db diabetes	[101]
	*Lactobacillus plantarum* HAC01	Mouse	HFD STZ-induced diabetes	[103]
	*Lactobacillus paracasei* NL41	Rat	HFD STZ-induced diabetes	[104]
	*Lactobacillus acidophilus* KLDS1.1003 and KLDS1.0901	Mouse	HFD STZ-induced diabetes	[110]
	SLAB51 composite probiotics	Mouse	AD	[111]
HOMA-IR improvement (reduction)	*Bifidobacterium animalis* 01	Rat	High-fat chow diet STZ-induced diabetes	[81]
	*Lactobacillus paracasei* HII01	Rat	HFD STZ-induced diabetes	[82]
	*Lactobacillus rhamnosus* (YC), *Bifidobacterium adolescentis* (N3) and *Bifidobacterium bifidum* (M2)	Mouse	HFD STZ-induced diabetes	[77]
	Heat-inactivated *Streptococcus* *thermophilus*	Rat	HFD ZDF diabetes	[84]
	*Bacillus licheniformis* Zhengchangsheng^®^	Mouse	HFD-induced obesity	[87]
	Probioglu^TM^	Rat	STZ-induced diabetes	[89]
	*Bacillus toyonensis* SAU-19	Mouse	HFD ZDF-induced diabetes	[90]
	*Lactobacillus casei* LC89	Mouse	STZ-induced diabetes	[91]
	Lactibiane Tolérance^®^	Mouse	HFD	[92]
	*Lactobacillus plantarum* CCFM0236	Mouse	STZ-induced diabetes	[78]
	*Lactobacillus plantarum* YJ7	Mouse	HFD	[94]
	*Lactobacillus plantarum* DSM 15313	Mouse	HFD-induced obesity	[97]
	14 composite probiotics	Mouse	Db/db diabetes	[100]
	14 composite probiotics	Mouse	Db/db diabetes	[101]
	*Lactobacillus plantarum* HAC01	Mouse	HFD STZ-induced diabetes	[103]
	*Lactobacillus paracasei* NL41	Rat	HFD STZ-induced diabetes	[104]
	VSL#3 composite probiotics	Mouse	HFD-induced obesity	[106]
	*Lactobacillus acidophilus* KLDS1.0901	Mouse	HFD STZ-induced diabetes	[110]
HOMA-B reduction	*Romboutsia ilealis* DSM 25109	Mouse	Western diet-induced diabetes	[112]
↓ fasting insulin levels	*Lactobacillus paracasei* HII01	Rat	HFD STZ-induced diabetes	[82]
	Heat-inactivated *Streptococcus* *thermophilus*	Rat	HFD ZDF diabetes	[84]
	*Bacillus licheniformis* Zhengchangsheng^®^	Mouse	HFD-induced obesity	[87]
	Probioglu^TM^	Rat	STZ-induced diabetes	[89]
	*Lactobacillus casei* LC89	Mouse	STZ-induced diabetes	[91]
	*Romboutsia ilealis* DSM 25109	Mouse	Western diet-induced diabetes	[112]
	Lactibiane Tolérance^®^	Mouse	HFD	[92]
	*Lactobacillus plantarum* CCFM0236	Mouse	STZ-induced diabetes	[78]
	*Lactobacillus acidophilus* KLDS1.0901	Mouse	HFD STZ-induced diabetes	[110]
↑ insulin levels	*Lactobacillus casei* and *Bifidobacterium bifidum* (alone or combination of both)	Rat	STZ-induced diabetes	[79]
	*Lactobacillus rhamnosus* (BSL and R23)	Rat	STZ-induced diabetes	[83]
	14 composite probiotics	Mouse	Db/db diabetes	[100]
Insulin sensitivity improvement (↑ ISI, QUICKI test or others)	*Lactobacillus rhmanosus* (GG, YC), *Bifidobacterium adolescentis* (7-2) and *Bifidobacterium bifidum* (35)	Mouse	HFD STZ-induced diabetes	[77]
	*Lactobacillus rhamnosus* (BSL and R23)	Rat	STZ-induced diabetes	[83]
	*Bacillus amyloliquefaciens* SC06	Mouse	HFD-induced obesity	[113]
	*Lactobacillus plantarum* CCFM0236	Mouse	STZ-induced diabetes	[78]
	*Lactobacillus plantarum* DSM 15313	Mouse	HFD-induced obesity	[97]
	*Lactobacillus acidophilus* KLDS1.0901	Mouse	HFD STZ-induced diabetes	[110]
Glucose tolerance improvement (OGTT)	*Bifidobacterium longum* APC1472	Mouse	HFD	[45]
	*Lactobacillus casei* and *Bifidobacterium bifidum* (alone or combination of both)	Rat	STZ-induced diabetes	[79]
	*Lactobacillus rhmanosus*, *Bifidobacterium adolescentis* and *Bifidobacterium bifidum* (all together 9 strains)	Mouse	HFD STZ-induced diabetes	[77]
	*Lactobacillus paracasei* HII01	Rat	HFD STZ-induced diabetes	[82]
	*Bifidobacterium animalis* 01	Rat	High-fat chow diet STZ-induced diabetes	[81]
	*Lactobacillus rhamnosus* (BSL and R23)	Rat	STZ-induced diabetes	[83]
	Heat-inactivated *Streptococcus thermophilus*	Rat	HFD ZDF diabetes	[84]
	*Weissella cibaria* MG5285, *Limosilactobacillus reuteri* MG5149, *Lacticaseibacillus rhamnosus* MG4502, *Lactobacillus gasseri* MG4524	Mouse	HFD-induced obesity	[88]
	*Romboutsia ilealis* DSM 25109	Mouse	Western diet-induced diabetes	[112]
	Probioglu^TM^	Rat	STZ-induced diabetes	[89]
	*Lactobacillus salivarus* AP-32 *Lactobacillus reuteri* GL-104	Mouse	Db/db obesity	[93]
	*Bifidobacterium longum* BR-108 (inactivated)	Mouse	Obese diabetes	[95]
	*L. rhamnosus* L12, *L. acidophilus*, *L. plantarum* HM218749, *B. animalis* subsp. *lactis* LPL-RH, *B. longum* subsp. *longum* BAMA-B05/BAu-B1024)	Mouse	Glucose STZ-induced diabetes	[96]
	*L. rhamnosus* L12, *L. acidophilus*, *L. plantarum* HM218749, *B. animalis* subsp. *lactis* LPL-RH, *B. longum* subsp. *longum* BAMA-B05/BAu-B1024)	Mouse	Glucose water- Induced diabetes	[96]
	*Lactobacillus rhamnosus* GG	Mouse	Db/db obesity	[114]
	*Bifidobacterium longum*	Mouse	HFD-induced obesity	[102]
	*Lactobacillus plantarum* HAC01	Mouse	HFD STZ-induced diabetes	[103]
	*Bifidobacterium longum* OLP-01	Mouse	HFD-induced obesity	[115]
	*Limosilactobacillus fermentum* MG4231 and MG4244	Mouse	HFD-induced obesity	[116]
	*Lactobacillus paracasei* NL41	Rat	HFD STZ-induced diabetes	[104]
	VSL#3 composite probiotics	Mouse	HFD-induced obesity	[106]
	*Lactobacillus acidophilus* SJLH001	Mouse	HFD	[108]
	*Lactobacillus casei* Zhang	Rat	Hyperinsulinemia	[109]
↑ osteocalcin	*Lactobacillus casei* Zhang	Rat	Hyperinsulinemia	[109]
↓ IL-6	Heat-inactivated *Streptococcus* *thermophilus*	Rat	HFD ZDF-induced diabetes	[84]
	Probioglu^TM^	Rat	STZ-induced diabetes	[89]
	*Lactobacillus plantarum* YJ7	Mouse	HFD-induced obesity	[94]
	*Bacillus amyloliquefaciens* SC06	Mouse	HFD-induced obesity	[113]
	*Lactobacillus casei* LC89	Mouse	STZ-induced diabetes	[91]
↓ TNF-α	*Lactobacillus plantarum* YJ7	Mouse	HFD-induced obesity	[94]
	*Bacillus amyloliquefaciens* SC06	Mouse	HFD-induced obesity	[113]
	Heat-inactivated *Streptococcus* *thermophilus*	Rat	HFD ZDF-induced diabetes	[84]
	Probioglu^TM^	Rat	STZ-induced diabetes	[89]
	*Lactobacillus casei* LC89	Mouse	STZ-induced diabetes	[91]
	SLAB51 composite probiotics	Mouse	AD	[117]
	*Bifidobacterium animalis* 01	Rat	High-fat chow diet STZ-induced diabetes	[81]
	*Lactobacillus plantarum* CCFM0236	Mouse	HFD STZ-induced diabetes	[78]
↓ IL-1β	*Lactobacillus casei* LC89	Mouse	STZ-induced diabetes	[91]
	*Lactobacillus plantarum* YJ7	Mouse	HFD-induced obesity	[94]
	Heat-inactivated *Streptococcus* *thermophilus*	Rat	HFD ZDF-induced diabetes	[84]
	Probioglu^TM^	Rat	STZ-induced diabetes	[89]
	SLAB51 composite probiotics	Mouse	AD	[117]
↑ IL-10	Heat-inactivated *Streptococcus* *thermophilus*	Rat	HFD ZDF-induced diabetes	[84]
	*Lactobacillus casei* LC89	Mouse	STZ-induced diabetes	[91]
	*Bifidobacterium animalis* 01	Rat	High-fat chow diet STZ-induced diabetes	[81]
	*Lactobacillus plantarum* CCFM0236	Mouse	HFD STZ-induced diabetes	[78]

T2D: type 2 diabetes; FPG: fasting plasma glucose; HbA1c: glycohemoglobin; HOMA-IR: homeostasis model assessment-estimated insulin resistance; HOMA-B: index reflecting pancreatic beta-cell function; ISI: insulin sensitivity index; QUICKI: quantitative insulin sensitivity check index; OGTT: oral glucose tolerance test; IL-6: interleukin-6; IL-10: interleukin 10; IL-1β: interleukin 1 beta; TNF-α: tumour necrosis factor alpha; ZDF: Zucker diabetic fatty; HFD: high-fat diet; STZ: streptozotocin; Db/db: genetically mutated model for type 2 diabetes with improper leptin receptor function; AD: Alzheimer’s disease; ↓: decrease; ↑: increase.

**Table 3 life-12-01187-t003:** Effect of probiotics on intestinal microbiota and glucose metabolism in in vivo animal model studies.

Animal Model/ Pathology	Administered Probiotic	Altered Intestinal Microbiota		Effect on Glucose Metabolism	Reference
		Genera or Family			
		Increase	Decrease		
T2D mouse	*Lactobacillus casei* LC89	*Alloprevotella* (B), *Bacteroides* (B), *Parabacteroides* (B), *Ruminococcus* (F)	*Lachnospiraceae* *_NK4A136_group* (F), *Odoribacter* (B) *Mucispirillum* (D)	Glucose tolerance improvement	[91]
HFD- induced obese mouse	*Lactobacillus* *plantarum* HT121	*Akkermansia* (V), *Allobaculum* (F), *Lactobacillus* (F), *Prevotella* (B), *Sutterella* (P) and S24-7 family (B)	*Anaerostipes* (F) *Anaerotruncus* (F) *Bilophila* (P) *Candidatus_Arthromitus* (F) *Coprococcus* (F), *Dorea* (F), *Mucispirillum* (D), *Oscillospira* (F), *Ruminococcus* (F), *Streptococcus* (F) and families of Peptococcaceae (F), Ruminococcaceae (F)	Glucose tolerance improvement; blood glucose decrease	[197]
HFD- induced obese mouse	*Lactobacillus* *plantarum* Q180		Rikenellaceae (B), Ruminococcaceae (F), Lachnospiraceae (F)	glucose tolerance improvement	[142]
T2D mouse	*Lactobacillus casei* CCFM419	*Allobaculum* (F), *Bacteroides* (B)		Alleviation of type 2 diabetes symptoms (insulin resistance and hyperglycaemia amelioration)	[198]
T2D mouse	*Lactobacillus* *plantarum* HAC01	Akkermansiaceae (V)	Dusulfovibrionaceae (P)	Reduction in glucose-mediated insulin secretion (SCFA increase)	[103]
T2D mouse (db/db)	14 composite probiotics	*Bifidobacterium* (A), *Lactobacillus* (F), *Clostridium leptum* (F), *Roseburia* (F), *Prevotella* (B),	*Enterococcus faecium* (F), *Escherichia coli* (P), *Bacteroides* *thetaiotaomicron* (B)	Improvement in glucose absorption	[101]
T2D rat	*Lactobacillus* G15 and Q14	*Clostridium leptum* (F), *Bacteroides* (B), *Prevotella* (B)	*Bifidobacterium* (A), *Lactobacillus* (F)	Improvement in blood glucose and insulin disorders (glucose tolerance)	[199]
T2D mouse	**^Bifidobacterium* *adolescentis* N3, **^Bifidobacterium* *bifidum* M2, *^Bifidobacterium* *adolescentis* 7-2, °*Lactobacillus* *rhamnosus* YC, ’*Lactobacillus* *rhamnosus* 7-1	** Bacteroidales* S24-7 (B), *^Parabacteroides* (B), °*Mucispirillum* (D), ’*Coprococcus* (F), ’*Streptococcus* (F)		Glucose metabolism and insulin resistance improvement (reduction in blood glucose levels and insulin resistance, regulation of SCFAs levels)	[77]
T2D mouse	*Lactobacillus* *acidophilus* KLDS1.003 and KLDS1.0901	*Blautia* (F), *Roseburia* (F), *Anaerotruncus* (F)	*Desulfovibrio* (T), *Alistipes* (B) *Bacteroides* (B)	Glucose and lipid metabolism- related signalling route improvement of intestinal microbiota	[110]
T2D rat	*Streptococcus* *thermophilus*	*Ruminococcaceae* (F), *Veillonella* (F), *Coprococcus* (F), *Barnesiella* (B)		Moderation of insulin resistance (HOMA-IR, HbA1c improvement)	[84]
T2D monkey	*Lactobacillus* *plantarum* - pMG36e-GLP-1	*Alistipes* (B)	*Prevotella* (B)	Blood glucose reduction (increase in SCFAs)	[200]
GDM pregnant rat	*Lactobacillus* *rhamnosus* LGG and *Bifidobacterium animalis* subsp. *lactis* Bb12	Lachnospiraceae (F), *Dubosiella* (F)	Muribaculaceae (F), *Ruminococcacea_* *UCG-005* (F)	Carbohydrate metabolism and membrane transport pathway inhibition	[201]

T2D: type 2 diabetes; HFD: high-fat diet; db/db: db/db mouse has a mutation of the diabetes (*db*) gene encoding for the *ObR*; GDM: gestational diabetes mellitus; F: Firmicutes (Bacillota); B: Bacteroidetes (Bacteroidota); V: Verrucomicrobia (Verrucomicrobiota); D: Deferribacteres (Deferribacterota); P: Proteobacteria (Pseudomonadota); A: Actinobacteria (Actinomycetota); T: Thermodesulfobacteriota; HOMA-IR: homeostasis model assessment-estimated insulin resistance; HbA1c: glycohemoglobin; SCFA: short-chain fatty acids; *, ^, °, ’: each probiotic affects different types of bacteria according to the same symbol.

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
