# Peer review of "Probiotic Mechanisms Affecting Glucose Homeostasis: A Scoping Review"

_life, 2022, doi:10.3390/life12081187_

Round 1
Reviewer 1 Report
In the present paper, the authors discussion the probiotic effects at different organ levels and their potential mechanisms to influence glucose metabolism and its homeostasis. Moreover,variations in results due to different methodological approaches were discussed, as well as limitations of the studies.They included both in vivo and in vitro studies in their article.
They discussed the effect of probiotics on the secretion of the intestinal hormones incretins which directly influence glucose metabolism and homeostasis. Effect of probiotics on permeability and integrity of the intestinal wall ,as well as reduction in both pro-inflammatory parameters which proved an improvement in intestinal restoration and insulin resistance was shown. The effect of probiotics on glucose transport in intestine showed a reduction in the intestinal glucose absorption .
Please find hereby some more articles which should of interest to your discussion
The Emerging Role of the Gut Microbiome in Cardiovascular Disease: Current Knowledge and Perspectives
Papadopoulos, P.D., Tsigalou, C., Valsamaki, P.N., ...Voidarou, C., Bezirtzoglou, E.,Maintaining digestive health in diabetes: The role of the gut microbiome and the challenge of functional foods
Bezirtzoglou, E., Stavropoulou, E., Kantartzi, K., ...Varzakas, T., Kourkoutas, Y.,Fermentative foods: Microbiology, biochemistry, potential human health benefits and public health issues
Voidarou, C., Antoniadou, M., Rozos, G., ...Lagiou, A., Bezirtzoglou, E. ,Probiotics in Medicine: A Long Debate
Stavropoulou, E., Bezirtzoglou, E. ,Potential Elimination of Human Gut Resistome by Exploiting the Benefits of Functional Foods
Tsigalou, C., Konstantinidis, T., Stavropoulou, E., Bezirtzoglou, E.E., Tsakris, A.Author Response
Dear Reviewer,
Thank you for your comments and suggestions on our manuscript (Ms. Life-1801344) entitled "PROBIOTIC MECHANISMS AFFECTING GLUCOSE HOMEOSTASIS: A SCOPING REVIEW". Please find below detailed responses to the your comments:
Please find hereby some more articles which should of interest to your discussion
Thank you for your comment and suggested articles. The provided articles are all reviews and we cited them at various points in the manuscript according to their content, wherever possible, to improve the manuscript. Please see below the respective places of additional articles cited in the manuscript:
The Emerging Role of the Gut Microbiome in Cardiovascular Disease: Current Knowledge and Perspectives
Papadopoulos, P.D., Tsigalou, C., Valsamaki, P.N., ...Voidarou, C., Bezirtzoglou, E.Biomedicines, 2022, 10(5), 948
(Page 2, 1. Introduction, 3 rd paragraph; Page 20, 3.8.1., 1 st paragraph).
Maintaining digestive health in diabetes: The role of the gut microbiome and the challenge of functional foods
Bezirtzoglou, E., Stavropoulou, E., Kantartzi, K., ...Varzakas, T., Kourkoutas, Y.Microorganisms, 2021, 9(3), pp. 1–26, 516
(Page 2, 1. Introduction, 3 rd paragraph;Page 21, 1 st paragraph).
Fermentative foods: Microbiology, biochemistry, potential human health benefits and public health issues
Voidarou, C., Antoniadou, M., Rozos, G., ...Lagiou, A., Bezirtzoglou, E. Foods, 2021, 10(1), 69
(Page 2, 1. Introduction, 3 rd paragraph)
Probiotics in Medicine: A Long Debate
Stavropoulou, E., Bezirtzoglou, E. Frontiers in Immunology, 2020, 11, 2192
(Page 12, 3.3., 1 st paragraph; page 18, 4 th paragraph; page 20, 3.8.1., 2 nd paragraph; page 21, 1 st paragraph)
Potential Elimination of Human Gut Resistome by Exploiting the Benefits of Functional Foods
Tsigalou, C., Konstantinidis, T., Stavropoulou, E., Bezirtzoglou, E.E., Tsakris, A. Frontiers in Microbiology, 2020, 11, 50
(Page 21, 4 th paragraph; 27-28, 4. Concluding remarks, 3 rd paragraph).
While it is a really interesting article to be published I think that discussion must be improved
Reviewer 2 Report
The authors performed a scoping review with the aim of exploring the main probiotic mechanism affecting glucose homeostasis. The topic has been preliminary discussed before. We can find several similar reviews published in recent years. The authors should address the significance of this manuscript compared with other previous reports. However, the article may be accepted after a major revision.
Major points:
- This reviewer was wondering whether there is a publication bias since the current review is not systematic. Author should pointed this statement as limitation;
- I suggest to start chapter 3 with subchapters 3.3-3.8, because as mentioned in your title and according to main purposes you try to discover and discussed the main probiotic mechanism affecting glucose homeostasis!!
- You don’t correctly interpreted results from study 142, in condition of HFD-induced obesity the level of adiponectin was decreased and restored after probiotic administration
- Moreover, it is already known that the activity of probiotics is species-specific, so I suggest to update the future perspectives with data related to effectiveness of novel probiotics strain such as Akkermansia muciniphila, Faecalibacterium prausnitzii, Prevotella copri etc.
Minor points:
- Page 3: in vitro and in vivo should be in Italics;
- In table 1 and 2 “glycaemic blood parameters” is not correct term. For example HOMA-IR and fasting insulin, hormones and cytokines which are not glycemia-related parameters. Also I don’t find in table 1 and 2 physiological condition (row 3). What differences between reduction of HOMA-IR (which is main indirect marker to asses insulin resistance) and insulin resistance reduction (improvement)?
- Page 5: “In another 12-week human clinical trial, the effect of probiotic on some parameters of glycaemic control, fat levels and total serum bile acids could not be demonstrated” – not clearly understandable
- endotoxic adipokines – what do you mean ?
- SIRTs (page 6), TGF-β (page 15) – please give the abbreviation after first meet in text
- Page 6 – “Despite the proven beneficial effects of probiotic therapy in patients with type 2 dia-betes, some studies failed to demonstrate changes” – which changes ?
- “on an empty stomach” – better use in fasting state or condition
- Page 11 - “glucose sensitivity” - better use in glucose tolerance
- Page 12 – “as well as synthesis of the body mass” - what do you mean ?
- Page 12 – in vitro studies – please extend this paragraph
- I don’t find probiotic in figure 1? As well in terms insulin sensitivity improvement adiponectin should be increased ?
- Page 14 – “linked to diseases such as insulin resistance” – insulin resistance is not disease
- Page 26 – “GLU5 expression” ?
- References # 47, 85, 87, 101, 113, 123, 206 and 207 are not within journal requirements; surnames for authors are abbreviated
Author Response
Dear Reviewer,
Thank you for sending us your comments and suggestions about our manuscript (Ms. Life-1801344) entitled "PROBIOTIC MECHANISMS AFFECTING GLUCOSE HOMEOSTASIS: A SCOPING REVIEW". Please find below detailed responses to your comments:
The authors performed a scoping review with the aim of exploring the main probiotic mechanism affecting glucose homeostasis. The topic has been preliminary discussed before. We can find several similar reviews published in recent years. The authors should address the significance of this manuscript compared with other previous reports. However, the article may be accepted after a major revision.
Thank you for your comment and for your concern. We have indeed pointed out the growing evidence of many articles/reviews dealing with the effects of probiotics on glucose metabolism and glucose homeostasis (third paragraph in the Introduction chapter) and we can agree with you on this point. Each of these reviews, however, points to specific areas of direct or indirect probiotic action (some are listed below):
- conditions and diseases (type 2 diabetes: DOI: 10.1080/10408398.2018.1547268; DOI: 10.1080/10408398.2018.1547268; https://doi.org/10.1016/j.fbio.2021.101172; DOI: 10.3390/nu14010166; obesity: DOI: 10.1016/j.foodres.2021.110490; DOI: 10.1155/2019/3291367; Cardiovascular-metabolic diseases: https://doi.org/10.1016/j.phanu.2021.100261
- organs/tissues (https://doi.org/10.1016/j.tifs.2021.04.040; doi:10.3389/fendo.2019.00029)
These reviews are rather concise, which is understandable, because the topic of glucose metabolism and homeostasis is really very popular, but on the other hand also very important in specific physiological and pathological aspects.
Indeed there are few reviews with a similar title to ours. One is from 2013 (DOI: 10.3109/09637486.2013.775227) and the other is a more recent from 2021 (DOI: 10.1021/acs.jafc.1c04291). The first discussed primary and secondary outcomes (various blood parameters and markers) from animal and human studies and concluded with some proposed mechanisms of action. The second focused only on human studies, on three types of probiotics, discussed their safety and listed the known probiotic functional foods that help control glucose homeostasis. They summarised only a limited number of mechanisms of how probiotic bacteria might regulate glucose homeostasis and discussed the emerging trend in probiotic functional foods.
Our aim was broader - to gather as much knowledge as possible about the interplay between probiotics and the maintenance of glucose homeostasis, in order to get a comprehensive picture of probiotic effects on all parts of the body under different pathological and/or physiological conditions. We believe that researchers, who are usually confined to specific research areas, can get new ideas for planning future work by reading a scoping review covering all contemporary knowledge in the field.
Major points:
- This reviewer was wondering whether there is a publication bias since the current review is not systematic. Author should pointed this statement as limitation;
Thank you for your comment. We understand your concern, but there was no publication bias between the articles meeting search strategy criteria listed in chapter 2.1. (Search strategy and studies's selection ). 137 articles were retrieved in total and they were all included in the review.
- I suggest to start chapter 3 with subchapters 3.3-3.8, because as mentioned in your title and according to main purposes you try to discover and discussed the main probiotic mechanism affecting glucose homeostasis!!
Thank you for your opinion. Internally we discussed pros and cons about keeping or leaving out these two chapters. Our conclusion was, that:
Chapter 3.1 describes general remarks about the mechanisms of probiotics which are indeed general, but still important for understanding the limitations of experiments with probiotics. In particular, less experienced researchers should be aware of them.
Chapter 3.2 describes the Effects of probiotics on blood parameters. The majority of in vivo clinical studies about the effects of probiotics on glucose homeostasis, followed various blood parameters to track the probiotic effect on the homeostasis. Essentially, blood examination is the most widely accepted and the least invasive method currently in practice. Therefore, a large number of blood parameters have been developed as a direct (e.g., insulin, glucose fasting levels etc.) or indirect measure of glucose homeostasis (e.g., cytokines, glycoalbumin etc.). We consider this chapter of prime importance to keep in the article, since the vast majority of data and results are at least partially based on easy-to-measure blood parameters.
Therefore, leaving out chapters 3.1 and 3.2 would significantly change the overall comprehensive evaluation of the topic. We suggest to keep it in the article.
- You don’t correctly interpreted results from study 142, in condition of HFD-induced obesity the level of adiponectin was decreased and restored after probiotic administration
Indeed, adiponectin levels decreased after LGG treatment and were restored – indicated by the statistically significant difference from the control. We corrected the statement as suggested (page 15).
- Moreover, it is already known that the activity of probiotics is species-specific, so I suggest to update the future perspectives with data related to effectiveness of novel probiotics strain such as Akkermansia muciniphila, Faecalibacterium prausnitzii, Prevotella copri etc.
We have included your suggestion in the chapter 4 (page 27; 3 rd paragraph).
Minor points:
- Page 3: in vitro and in vivo should be in Italics;
We have corrected as suggested and additionally unified the whole manuscript.
- In table 1 and 2 “glycaemic blood parameters” is not correct term. For example HOMA-IR and fasting insulin, hormones and cytokines which are not glycemia-related parameters. Also I don’t find in table 1 and 2 physiological condition (row 3). What differences between reduction of HOMA-IR (which is main indirect marker to asses insulin resistance) and insulin resistance reduction (improvement)?
Thank you for all your comments. Please find below our responses to each of them.
In table 1 and 2 “glycaemic blood parameters” is not correct term. For example HOMA-IR and fasting insulin, hormones and cytokines which are not glycemia-related parameters.
We partially agree. The word glycaemic has been omitted in both tables and in the title of subchapter 3.2. However, we have divided blood parameters into those that can be directly related to glycaemic status and those that have an indirect relationship. The underlying mechanisms about how blood parameters (such as hormones, cytokines, glycoproteins and others) can be indirectly related to the glucose status in blood, was discussed based on the original article author speculations / suggestions (Subchapter 3.2.).
Also I don’t find in table 1 and 2 physiological condition (row 3)
Both Table 1 and Table 3 have a column titled ‘’Physiological/Pathological condition’’. It contains information about the health status of patients / laboratory animals included in the study before probiotic treatment (physiological condition – healthy; pathological condition – pre-illness state/disease).
What differences between reduction of HOMA-IR (which is main indirect marker to asses insulin resistance) and insulin resistance reduction (improvement)?
HOMA-IR is a measurable parameter, while insulin resistance is a condition that can be reflected (measured) by different parameters. To avoid confusion, we have decided to include only blood parameters in the tables and omit conditions.
- Page 5: “In another 12-week human clinical trial, the effect of probiotic on some parameters of glycaemic control, fat levels and total serum bile acids could not be demonstrated” – not clearly understandable
We have rewritten the sentence to make it more understandable: “In another 12-week human clinical trial, the probiotic did not alter HbA1c levels, fat levels or total serum bile acids.”.
- endotoxic adipokines – what do you mean ?
Thank you for noticing an obvious typo. The correct word is endotoxin and it has been corrected.
- SIRTs (page 6), TGF-β (page 15) – please give the abbreviation after first meet in text
We have corrected as suggested.
- Page 6 – “Despite the proven beneficial effects of probiotic therapy in patients with type 2 dia-betes, some studies failed to demonstrate changes” – which changes ?
We have added “in blood and metabolic parameters”.
- “on an empty stomach” – better use in fasting state or condition
We corrected and unified the term as suggested.
- Page 11 - “glucose sensitivity” - better use in glucose tolerance
We agree. Glucose sensitivity was explained as glucose tolerance when first used in the text. The term ''glucose tolerance'' was then used throughout the manuscript.
- Page 12 – “as well as synthesis of the body mass” - what do you mean ?
We meant as ‘’synthesis of the muscle body mass’’. To exclude any misunderstanding we have corrected in: “as well as synthesis of the muscle mass”.
- Page 12 – in vitro studies – please extend this paragraph
We agree. We have corrected as suggested.
- I don’t find probiotic in figure 1? As well in terms insulin sensitivity improvement adiponectin should be increased ?
I don’t find probiotic in figure 1?
In the figures we wanted to present the main results, i.e., the probiotic effects themselves, their endpoint effects or consequences. Therefore, in our opinion it is not important to include a picture of the probiotic in these figures to keep them as simple as possible. Furthermore, the term probiotic is mentioned in all Figure captions.
As well in terms insulin sensitivity improvement adiponectin should be increased ?
We absolutely agree. We have corrected the mistake (arrow) in Figure 1 according to the text.
- Page 14 – “linked to diseases such as insulin resistance” – insulin resistance is not disease
We agree. It is more a pathological condition than a disease. We have omitted the word and corrected the mistake.
- Page 26 – “GLU5 expression” ?
We have corrected the mistake (GLUT5).
- References # 47, 85, 87, 101, 113, 123, 206 and 207 are not within journal requirements; surnames for authors are abbreviated
We have corrected the mistakes.
Reviewer 3 Report
This review lists a great many recent publications where reference has been made to links between probiotic supplementation and control of glucose homeostasis. It is an interesting topic and one which has current interest for researchers and lay readers.
Major criticisms
I enjoyed reading this review but my main criticism is that it is basically a list. It draws on a great many publications and refers to many studies with their reported impacts of probiotics on glucose homeostasis. This, in itself, constitutes valuable reference material. However, the review would have far greater impact were more effort made to give a unifying view of the meaning of microbiotal composition on glucose metabolism. As it is, we read a great many observations on this subject but have little means of combining them. I believe it is within the authors’ capabilities to attempt to unify these observations.
On a similar note, it is often difficult to regard one aspect of the metabolic network to the exclusion of others. Many of the reported effects of probiotics on glucose homeostasis may be interpreted by the linking roles of ketone bodies, whether as energy-yielding substrates or as signalling molecules. I was surprised that this aspect was not mentioned during the review and I believe it should be. It follows from the interplay between intestinal microbial metabolism and the dependent metabolism of host intestinal epithelial cells which has attracted recent research attention and could be expanded in the current review.
Partly, these shortcomings arise from the authors’ approach in aiming to included every publication on the general topic of glucose homeostasis and the microbiome. Perhaps they might consider focusing on a smaller subset of effects and giving full mechanistic evaluations.
I would also have liked to see some commentary on clinical approaches to manipulation of the intestinal microbiome, not just via oral supplementation but via faecal transplantation and recorded impacts on IR, etc.
Minor criticisms
The manuscript is very well written, articulate and easy to understand. There are a number of typos which really should be corrected.
Author Response
Dear Reviewer,
Thank you for sending us your comments and suggestions about our manuscript (Ms. Life-1801344) entitled "PROBIOTIC MECHANISMS AFFECTING GLUCOSE HOMEOSTASIS: A SCOPING REVIEW". Please find below detailed responses to your comments:
This review lists a great many recent publications where reference has been made to links between probiotic supplementation and control of glucose homeostasis. It is an interesting topic and one which has current interest for researchers and lay readers.
Major criticisms
I enjoyed reading this review but my main criticism is that it is basically a list. It draws on a great many publications and refers to many studies with their reported impacts of probiotics on glucose homeostasis. This, in itself, constitutes valuable reference material.
Thank you for your comments and concerns. Our scoping review is structured according to inclusion and exclusion criteria and contains studies from a large time period (11 years) with a rather important topic. Therefore, we found many more articles than in a normally structured review. However, our aim was to show all the aspects studied (from all angles) of the administration of probiotics and their effects on the host in terms of affecting potential organs related to glucose metabolism and homeostasis. Therefore, we decided to analyse the effects of probiotics from these studies and present them separately according to different domains. We have tried to link their effects with glucose homeostasis from different angles (i.e. in different chapters of the review), but according to the authors' presentations/analyses/suggestions/speculations. We believe that in this way a much more comprehensive picture of the effects of probiotics on health and disease could emerge and new ideas could be realised in the future.
However, the review would have far greater impact were more effort made to give a unifying view of the meaning of microbiotal composition on glucose metabolism. As it is, we read a great many observations on this subject but have little means of combining them. I believe it is within the authors’ capabilities to attempt to unify these observations.
Thank you for your comments and suggestions. We agree with you that the link between probiotics, microbiotal composition and glucose metabolism is of great importance when studying the effects of probiotics on the host. Indeed, we have included an entire subchapter (3.8.1. Effect of probiotics on the intestinal microbiota; pages 19-24) in the review with a Table 3 (Animal models) showing some associations between the modulated microbiota after probiotic administration and glucose metabolism (Table 3, 4th column).
We are aware that probiotics act on multiple levels. Therefore, we have tried to improve this subchapter (and the table - 4th column) as much as possible (all the information we could obtain from the selected studies on the relationship between probiotics and microbiota and glucose). Accordingly, we have also improved the last chapter 4. - Concluding remarks and future perspectives (page 28, 2nd paragraph) with a brief summary about the observations related to gut microbiota. It should be noted, however, that standardisation along the lines of "one model fits all" is hardly possible due to the enormous species and strain specificity of probiotics. Therefore, we have chosen to divide the studies into as many chapters (consequences of probiotic action), more or less according to the effects of the probiotic mechanism.
On a similar note, it is often difficult to regard one aspect of the metabolic network to the exclusion of others. Many of the reported effects of probiotics on glucose homeostasis may be interpreted by the linking roles of ketone bodies, whether as energy-yielding substrates or as signalling molecules. I was surprised that this aspect was not mentioned during the review and I believe it should be. It follows from the interplay between intestinal microbial metabolism and the dependent metabolism of host intestinal epithelial cells which has attracted recent research attention and could be expanded in the current review.
Thank you for your comment and suggestion. Indeed, we believe that analysing the concentration of ketone bodies in the context of glucose and insulin metabolism may be of interest, as little is yet known about the function of ketones in hyperglycaemic states. Nevertheless, we thoroughly reviewed all selected studies to see if an analysis of ketone bodies was performed to analyse possible relationships between probiotics, ketone bodies and glucose. Among the 137 selected studies, we found only one animal model study (doi:10.1017/S0022029908003129) in which the authors measured ketone bodies in urine after administration of Lactobacillus acidophilus NCDC14 and Lactobacillus casei NCDC19. However, no ketone bodies were detected, so there was no discussion of possible correlations. It is hard to explain as to why all the other studies did not include ketone bodies in their investigations, other than speculation that they did not think they were important or simply did not include them in the analysis. Notwithstanding this fact, we decided to mention this aspect in the last chapter 4, penultimate paragraph (future perspectives).
Partly, these shortcomings arise from the authors’ approach in aiming to included every publication on the general topic of glucose homeostasis and the microbiome. Perhaps they might consider focusing on a smaller subset of effects and giving full mechanistic evaluations.
Thank you very much for your comment. Our aim was to gather as much knowledge as possible on this specific topic (effect of probiotics on the human/animal body in direct or indirect relation to glucose metabolism and homeostasis) in one place, in order to get a more comprehensive picture of probiotic effects on all parts of the body under different pathological and/or physiological conditions. We believe that in this way more possible connections about the effects of probiotics on health and disease can be explored in the future.
We understand that you would like to highlight the link between glucose homeostasis and the microbiome, which we also believe is very important. However, there was no publication bias between the articles as all the criteria listed in chapter 2.1. (Search strategy and selection of studies) were met in the 137 articles and therefore all were selected for review. We followed the rules of inclusion and exclusion criteria and extracted every possible link between the probiotic and its effect at all levels of the body and followed its effects on glucose metabolism and homeostasis.
We agree that a full mechanistic evaluation in fewer studies could contribute to a better understanding of some mechanisms underlying the probiotic mode of action. However, our aim was not to investigate the mechanisms themselves, but merely to suggest or highlight them as a possible consequence of various probiotic effects, exactly as assessed by the authors in the 137 included articles.
I would also have liked to see some commentary on clinical approaches to manipulation of the intestinal microbiome, not just via oral supplementation but via faecal transplantation and recorded impacts on IR, etc.
Thank you for your comment and suggestion. As your suggestion is not directly related to the scope of our review and we could not implement it in the main text, we have added it in the 4th chapter, last paragraph (future perspectives). We also believe that studies on new strategies to modulate the microbiota in relation to glucose metabolism and its homeostasis are of utmost importance for future research in this important field.
Minor criticisms
The manuscript is very well written, articulate and easy to understand. There are a number of typos which really should be corrected.
We have thoroughly checked and corrected typing errors.
Round 2
Reviewer 2 Report
The authors have satisfactorily addressed the concerns raised in the original version. The revised version is significantly improved. No further concerns.